# *CyberJurors*: A Multi-Agent Simulation Task for E-Commerce Disputes Verdict

**Yanhui Sun** [1]   **Wu Liu** [1]   **Haifeng Ming** [1]   **Xinru Wang** [1]   **Hantao Yao** [1]   **Yongdong Zhang** [1]

## Abstract

E-commerce platforms have begun recruiting crowdsourced jurors to adjudicate massive volumes of transaction disputes. Unlike formal legal judgment, E-commerce dispute verdicts require grounding pivotal clues from redundant, multi-round, multimodal evidence and making decisions under flexible platform-specific conventions. These characteristics render existing methods insufficient for this scenario. To bridge this gap, we introduce a pioneering task, *E-commerce Dispute Verdicts* (EDV), and present *VerdictBench*[1], a multimodal benchmark comprising 6,000 real-world cases designed to reflect crowdsourced jury decisions. Building upon this, we propose *CyberJurors*[2], a multi-agent framework to clarify the dispute logic and regulate the verdict process. At the individual level, *Individual Verdict Chain-of-Thought* decomposes the EDV task into four structured reasoning stages, enabling fine-grained clue perception and clarifying causal logic between pivotal clues and the dispute focus. At the collective level, *Jury Consensus Verdict* simulates multi-round discussion and voting among jurors, while incorporating verdict precedents to mitigate cognitive biases toward either disputant. Experiments on *VerdictBench* show that *CyberJurors* outperforms state-of-the-art LLMs, MLLMs, and court simulators, while achieving stronger alignment with real-world jury voting patterns.

## 1. Introduction

The rapid growth of the digital economy has led to an exponential increase in online transaction disputes. Traditional

[1] School of Information Science and Technology, University of Science and Technology of China, Hefei, China. Correspondence to: Wu Liu <liuwu@ustc.edu.cn>.

*Proceedings of the 43rd International Conference on Machine Learning*, Seoul, South Korea. PMLR 306, 2026. Copyright 2026 by the author(s).

[1] https://huggingface.co/datasets/piggi/VerdictBench

[2] https://github.com/YanhuiS/CyberJurors

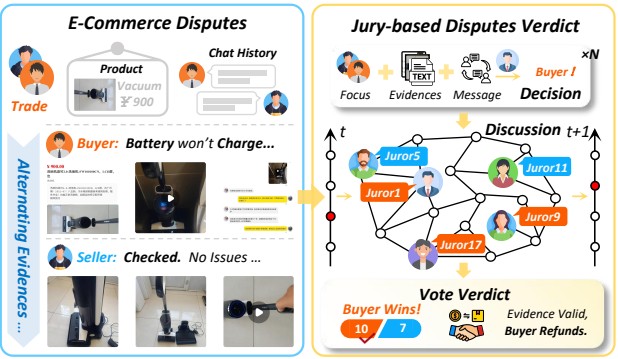

Figure 1. Illustration of the "E-commerce Dispute Verdicts" Task. The jury conducts an in-depth discussion of multimodal evidence alternately provided by disputants, capturing the dispute focus to achieve an accurate and fair verdict.

avenues for dispute resolution, such as judicial intervention (Fuller, 2018) and customer service (Barari et al., 2020), are inadequate to handle such voluminous demands. To improve efficiency, E-commerce platforms have introduced the "crowdsourced jury" mechanism. Specifically, the buyer and seller (*i.e.,* disputants) alternately submit multimodal evidence, including chat histories, images, and videos, to support their claims. Based on the evidence and their individual assessments, a panel of jurors then renders a verdict, and the side receiving the majority of votes is declared the winner, as illustrated in Fig. 1. However, constrained by their fragmented time, recruiting these volunteer jurors for each dispute typically takes several days, creating severe scalability bottlenecks for real-world deployment. In this context, developing intelligent systems for dispute verdicts has become an urgent practical need.

However, the distinctive characteristics of E-commerce disputes make existing dispute-related tasks difficult to transfer directly to the setting. As a representative paradigm, conventional legal judgment differs fundamentally from the E-commerce dispute verdicts in three dimensions: *verdict mechanisms*, *evidence structures*, and *guiding norms* (Fang et al., 2025; Chu et al., 2025; He et al., 2026). Specifically, the latter relies on crowdsourced jury decisions rather than professional legal adjudication (Talebirad & Nadiri, 2023; Leimeister, 2010). Moreover, it necessitates reasoning over redundant, multi-round, multimodal evidence instead of

structured textual evidence, and is governed by flexible, platform-specific conventions rather than rigid legal regulations. Motivated by this gap, we introduce a pioneering task, *E-commerce Dispute Verdicts* (EDV), which aims to model the fine-grained perception of multimodal evidence and replicate jury decision-making dynamics to achieve reliable, automated verdicts.

Recently, multi-agent systems have excelled in optimizing complex problems from multiple perspectives, providing fundamental support for intelligent dispute verdicts (Liu et al., 2026; Hong et al., 2024; Yuan & Xie, 2025; Du et al., 2024). For instance, AgentCourt constructs adversarial reasoning among simulated legal roles, achieving precise judgments on structured legal documents (Chen et al., 2025a). However, applying such methods to the EDV task faces two major bottlenecks: First, E-commerce disputes involve fragmented evidence across multi-round interactions between disputants, including questioning, rebuttal, and clarification. Consequently, **pivotal clues are often buried within redundant evidence, obscuring the underlying causal logic.** For instance, the charging dispute in Fig. 1 necessitates deep video analysis to identify pivotal visual clues (*i.e.,* the 2% battery level and flashing indicator light). Existing methods, however, remain largely confined to textual reasoning (Westermann et al., 2023; Yue et al., 2025; Chen et al., 2025b). Even with MLLMs (Liu et al., 2025a), their passive, one-shot inference fails to perceive fine-grained clues from redundant evidence, or clarify the causal logic behind the dispute. This also explains the empirical phenomenon where existing MLLMs underperform compared to text-only LLMs on the EDV task. Second, **Unlike formal legal trials, EDV heavily relies on flexible, platform-specific transaction rules, and lacks clear guidelines to regulate the verdicts.** As shown in Fig. 1, when a seller's proof before purchase conflicts with a buyer's claim of immediate defect upon receipt, the jury supports the buyer based on prevailing transaction rules. In such flexible scenarios, existing models often reflect the intrinsic biases of the underlying generative models (Bender et al., 2021; Xu et al., 2024; Taubenfeld et al., 2024), compromising fairness and explainability. This limitation prevents current systems from harnessing the benefits of collective consensus simulation (Bjornsdottir et al., 2022).

To facilitate a robust evaluation of EDV, we introduce *VerdictBench*, the first Multimodal Disputes Verdicts Benchmark for E-commerce, comprising 6,000 real-world cases meticulously collected from publicly available data. Each case encompasses the transaction logs, multi-round multimodal evidence from disputants, and the ground-truth 17-juror verdict outcome. Based on this, we propose *CyberJurors*, a framework that integrates *Individual Verdict Chain-of-Thought* (IV-CoT) and *Jury Consensus Verdict* (JCV) to clarify the dispute logic and support a fair verdict process.

Specifically, *for an individual juror*, IV-CoT decomposes the EDV task into a structured four-stage reasoning process that covers understanding, perception, analysis, and verdict. It allows jurors to perceive fine-grained evidence clues and clarify the causal logic between clues and the dispute focus, thereby enhancing accuracy and interpretability. *For the collective jury*, drawing inspiration from "*Stare Decisis*" (Landes & Posner, 2013; Llewellyn, 2013), JCV simulates jury discussions guided by Verdict Precedents, and aggregates the decisions from heterogeneous jurors into a collective verdict summary. This design provides individual jurors with explicit guidelines and global references, thereby mitigating individual biases. Ultimately, the verdict outcome is determined through a majority vote of the jury, with the summary serving as an interpretable justification.

Extensive experiments on *VerdictBench* show that *CyberJurors* outperforms existing LLMs, MLLMs, and LLM-based court simulators, achieving accuracy improvements of 9.48%, 9.38%, and 6.19%, respectively. To summarize, we present the following contributions:

- We introduce a pioneering task, *E-commerce Dispute Verdict*, which aims to replicate jury decision-making through fine-grained perception of multimodal evidence chains, facilitating intelligent and reliable verdicts.
- We collect a high-quality multimodal E-commerce dispute dataset, *VerdictBench*, which preserves authentic evidence structures and advances research in content understanding, intent reasoning, and multi-agent simulation.
- We propose *CyberJurors*, integrating the Individual Verdict Chain-of-Thought with Jury Consensus Verdict, to achieve fine-grained clue perception, dispute-focused causal reasoning, and fair dispute verdicts.

**Conflict of Interest Disclosure** The authors declare no conflicts of interest. This work is not associated with any commercial product developed by the authors' employers.

## 2. Related Work

**Jury Simulation,** which prompts LLMs to emulate human jurors, has been widely adopted in dispute judgment (Liu et al., 2025b; Wang et al., 2025). A pioneering effort, ChatEval (Chan et al., 2024), simulates human-like discussion by constructing autonomous debate teams, facilitating logical reasoning and intelligent decisions for complex scenarios. This paradigm has since been extended to the legal domain, demonstrating substantial potential for judicial intelligence (He et al., 2024; Zhang et al., 2025). For instance, Agent-Court integrates adversarial evolvable strategies to perform dynamic knowledge learning, enhancing the legal reasoning abilities of jurors (Chen et al., 2025a). However, unlike formal court trials, EDV necessitates intricate causal reasoning from noisy multimodal evidence. Moreover, the individual biases (Xu et al., 2024; Wataoka et al., 2024) stemming

from the absence of clear verdict rules make such court simulators unsuitable for EDV.

**Dispute Datasets.** EDV requires verifying the authenticity of *Multi-Round*, *Multi-Modal* evidence, and reasoning about the causal relation between evidence and the dispute focus to generate fair and accurate verdicts. To highlight these characteristics, Table 1 compares representative dispute-related datasets. Existing legal-domain datasets, such as SimuCourt (He et al., 2024), LawBench (Fei et al., 2024b), CourtBench (Chen et al., 2025a), and LAiW (Dai et al., 2025), primarily consist of legal documents and verdict outcomes. They are inherently unimodal and single-round, diverging significantly from the multi-modal, multi-round evidence chains of E-commerce disputes. Moreover, ECOD relies on single-round text-image conversations between disputants to decide whether to display the buyer's feedback in the takeaway scenario (Chen et al., 2023). It lacks the multi-round adversarial interaction and remains closed-source, which limits its applicability to broader EDV scenarios.

## 3. VerdictBench

The lack of dispute datasets severely hampers the reliable evaluation and generalization analysis of EDV. To bridge this gap, we introduce *VerdictBench*, the first large-scale Multimodal E-commerce Dispute Verdicts dataset, preserving the authentic logical structure of E-commerce disputes.

### 3.1. Dataset Construction

To faithfully replicate the real-world dispute process, we develop a dataset construction pipeline, as depicted in Fig. 2.

**Data Collection.** On the E-commerce platform, each dispute case is reviewed by 17 jurors, after which the complete evidence chains and final verdict outcomes are visible to all participants. As shown in Fig. 1, it encompasses transaction metadata, chat histories, and multi-round multimodal evidence (text, images, and videos) from disputants, together with the ground-truth 17-juror verdict outcomes. Moreover, to mitigate noise and enhance semantic density, we apply multi-layered quality checks for each case: *(i)* verify structural consistency; *(ii)* audit the existence of visual evidence and its alignment with textual claims; and *(iii)* assign product-category labels and generate visual summaries for images/videos without textual descriptions. Next, all sensitive fields are strictly anonymized to protect disputants' privacy. With approval from our Academic Committee, we collect these authorized cases to construct the multimodal dispute dataset named *VerdictBench*. After being converted into JSON format and deduplicated, *VerdictBench* ultimately contains 6,000 high-quality cases across five common E-commerce categories, preserving the authentic multi-round interactions to support human-like reasoning.

Table 1. Comparisons of Different Datasets.

| Dataset | Modality | Multi-Round | Scale | Domain |
|---|---|---|---|---|
| SimuCourt, *EMNLP24* | T | ✗ | 420 | *Legal* |
| LawBench, *EMNLP24* | T | ✗ | 10,000 | *Legal* |
| CourtBench, *ACL25* | T | ✗ | 1,124 | *Legal* |
| LAiW, *COLING25* | T | ✗ | 11,000 | *Legal* |
| ECOD, *NLPCC23* | T,I | ✗ | 6,336 | *TakeAway* |
| VerdictBench | **T, I, V** | ✓(8) | **6,000 × 8** | **E-commerce** |

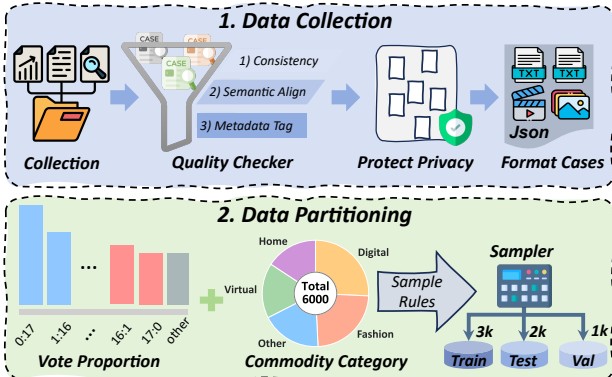

Figure 2. Dataset Construction Pipeline.

**Data Partitioning.** Statistics in Fig. 3(d) reveal that final verdicts and difficulty levels (real-world vote margin of the 17 jurors) vary across product categories. To ensure a comprehensive and robust evaluation, we adopt a stratified partitioning strategy based on category and difficulty with a 3:1:2 ratio, yielding balanced train, validation, and test sets. The strategy enables a more rigorous assessment of robustness and generalization in complex EDV scenarios.

### 3.2. Dataset Statistics

We conduct a multi-dimensional statistical analysis to characterize the inherent complexity of real-world EDV, as shown in Fig. 3.

**Overall Distribution.** *VerdictBench* covers five primary E-commerce categories (Fig. 3(a)), reflecting broad scenario coverage. Fig. 3(b) shows that, sellers win 62.6% of cases, compared to 37.4% for buyers. A plausible explanation is that sellers, as professional merchants, possess a deeper understanding of fulfillment conventions, thus holding more verifiable evidence. Such imbalance is more pronounced across categories (Fig. 3(c)). Furthermore, Fig. 3(d) illustrates the voting outcomes of the 17 jurors, revealing that verdict difficulties vary across categories. Refer to Appendix A.2 for statistics descriptions and case samples.

**Evidence Intensity.** Multimodal evidence is the cornerstone of EDV. Fig. 3(e) shows that each case averages 14 images and 0.9 videos with varying durations. Notably, buyers provide more than twice the amount of evidence compared to sellers (Fig. 3(f)), as they are more inclined to strengthen their claims through more evidence. However, evidence

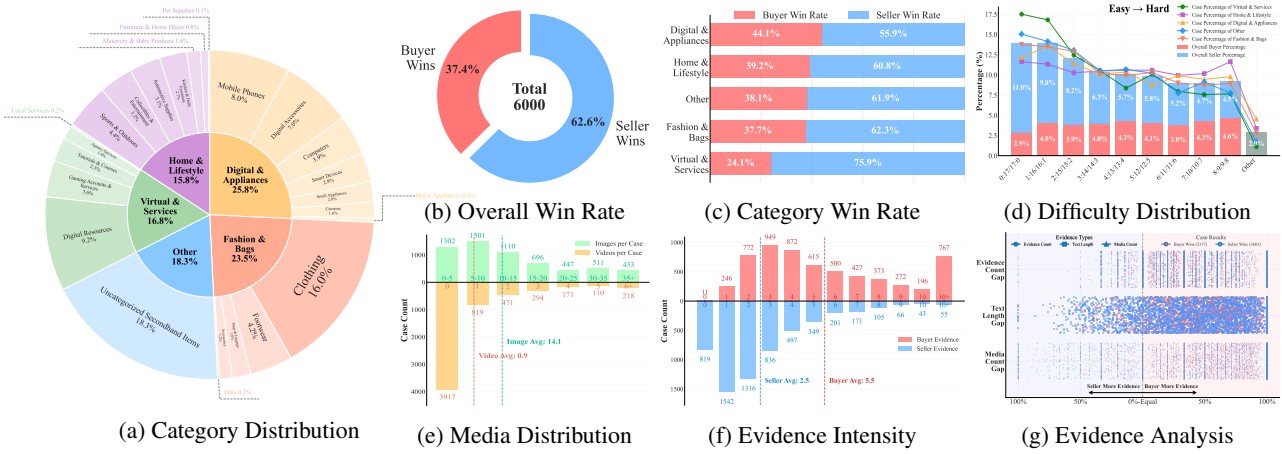

Figure 3. Visualization of Dataset Statistics. The figure illustrates the overall distribution (a-d) and evidence intensity (e-g) of *VerdictBench*.

intensity (rounds, text length, and media counts) does not necessarily translate into a verdict advantage (Fig. 3(g)). Winning a dispute hinges on whether cross-modal evidence accurately verifies the authenticity of the core claims from disputants. For instance, the buyer in Fig. 1 provides a video showing a flashing indicator light to align pivotal clues with the "charging" focus, shaping the jury's final verdict. Together with these insights, *VerdictBench* poses substantial technical challenges in fine-grained semantic understanding and logical reasoning, facilitating research on intent recognition and multi-agent simulation.

## 4. CyberJurors

Building upon *VerdictBench*, we propose *CyberJurors*, a framework that achieves fair and accurate dispute verdicts by integrating the Jury Consensus Verdict (JCV) with Individual Verdict Chain-of-Thought (IV-CoT), as depicted in Fig. 4. *At the collective level*, JCV emulates a multi-round jury discussion process among jurors guided by a Verdict Precedent Base, which provides explicit normative guidance and decision references for individual jurors. *At the individual level*, *i.e.,* for each juror in JCV, IV-CoT decomposes EDV into four stages: focus extraction, clues grounding, adversarial analysis, and final verdict, thereby clarifying the causal logic between pivotal clues and the dispute focus.

Formally, *CyberJurors* is modeled as a multi-agent system, characterized by a directed social network $G = \langle A, E \rangle$. Here, $A = \{a_1, a_2, ..., a_N\}$ represents a jury panel of $N$ heterogeneous jurors, and the directed edge $e_{k,j} \in E$ signifies that juror $a_k$ follows juror $a_j$. A dispute case, denoted as $D = \{d, e_1^b, e_1^s, \cdots, e_{N_b}^b, e_{N_s}^s\}$, contains transaction metadata $d$ (*e.g.*, product information, chat histories), $N_b$ pieces of buyer evidence $e_i^b$, and $N_s$ pieces of seller evidence $e_i^s$. Buyer evidence $e_i^b = \{T_i^b, I_i^b, V_i^b\}$ contains textual claims $T_i^b$, images $I_i^b$, and videos $V_i^b$. $e_i^s$ is defined analogously for the seller side.

For a given dispute case $D$, *CyberJurors* uses JCV to simulate $T$ rounds of interactive jury discussions. At each round, JCV provides all jurors with a Collective Verdict Summary and Verdict Precedent Base, assisting individual jurors in mitigating cognitive biases and adhering to normative guidelines. Based on the above context provided by JCV, individual juror $a_k$ uses IV-CoT to generate its verdict $\hat{y}_{k,t}$ and reason $J_{k,t}$. Specifically, juror $a_k$ parses textual claims to identify the dispute focus. Using the focus as a query, juror $a_k$ iteratively selects and perceives the most relevant visual content, precisely grounding pivotal clues. Subsequently, juror $a_k$ adversarially analyzes these pivotal clues from the disputants to uncover the causal logic between these clues and the dispute focus. By combining the contexts from JCV, juror $a_k$ then provides a final decision that is logically transparent and traceable to the evidence. At the end of each round, JCV aggregates the verdicts of all jurors and updates the collective verdict summary, which serves as a reference for the next round of jury discussion. After the final round, JCV derives the verdict based on the majority vote and outputs the collective verdict summary as an interpretable justification.

### 4.1. Individual Verdict Chain-of-Thought

During the jury simulation process, juror $a_k$ is required to reach a verdict for the dispute case $D$. This process essentially involves continuously grounding pivotal clues relevant to the dispute focus from redundant evidence, utilizing textual claims as indices to verify the correctness of the disputants' claims. However, existing methods typically leverage the passive perception capability of MLLMs, and process the entire dispute content in a one-shot manner. Such coarse-grained comprehension hinders jurors from performing the intricate "evidence-focus" causal reasoning. Motivated by the Chain-of-Thought (CoT) (Wei et al., 2022; Fei et al., 2024a), we propose the Individual Verdict Chain-of-Thought. It decomposes EDV into a four-stage structured

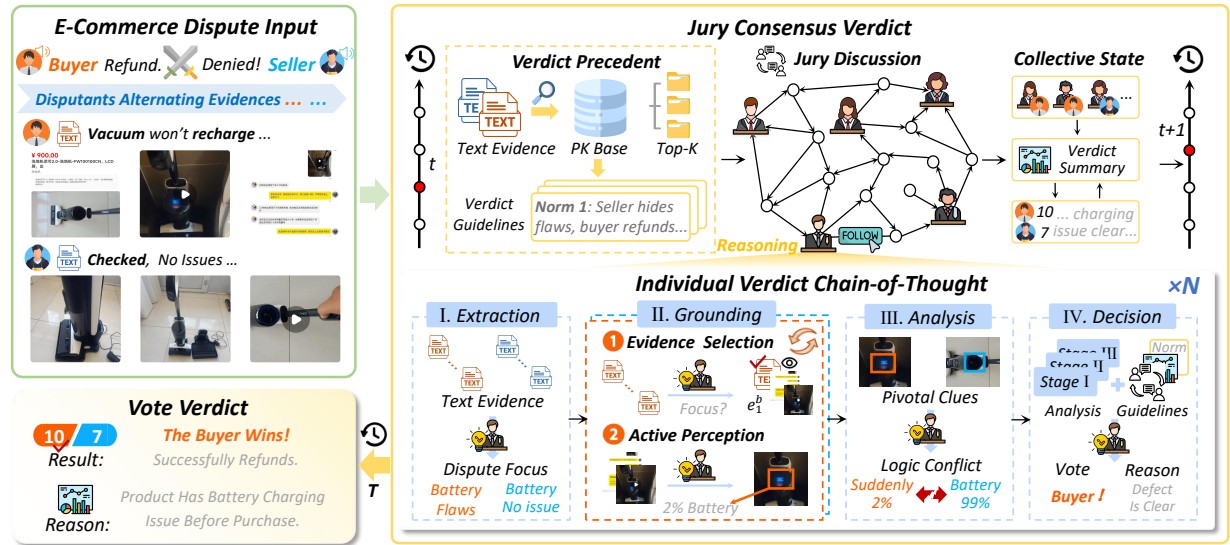

Figure 4. An Illustration of *CyberJurors*. (1) *Collective Level*: JCV emulates iterative jury discussions using Verdict Precedents as a normative reference. (2) *Individual Level*: Each juror performs reasoning via IV-CoT to clarify causal logic of disputes.

reasoning chain, transforming the reasoning paradigm from *"coarse-grained content encoding"* to *"fine-grained evidence grounding and causal inference"*.

▶ **Stage I: Focus Extraction.** Juror $a_k$ first performs structured parsing on the transaction metadata $d$ and the textual claims from disputants, to identify the dispute focus $F$ and extract the core demands of the disputants, $F^b$ and $F^s$,

$$O_{\mathrm{I}} : \{F, F^b, F^s\} = \mathcal{F}_{\mathrm{extract}}(d, \boldsymbol{T}^b, \boldsymbol{T}^s). \quad (1)$$

Here, $\boldsymbol{T}^b = \bigcup_{i=1}^{N_b} \boldsymbol{T}_i^b$ denotes all textual claims provided by the buyer, and $\boldsymbol{T}^s$ is defined similarly for the seller.

▶ **Stage II: Clues Grounding.** To facilitate an explicit causal mapping between pivotal clues and the dispute focus, we design a **"Select-Perceive" Iterative Clues Grounding** strategy. It transforms conventional one-shot multimodal comprehension into an active iterative process, which grounds pivotal clues to substantiate the disputants' claims. To avoid interference from mixing evidence, this strategy is performed independently for each disputant. For the buyer side at round $t$,

*1) Juror actively selects the evidence most likely to contain pivotal clues from the redundant evidence.* Given the dispute focus and the textual claims, the juror selects a piece of evidence $\boldsymbol{e}_{*,t}^b$ to verify the buyer's claims,

$$\boldsymbol{e}_{*,t}^b = \mathcal{F}_{\mathrm{select}}(O_{\mathrm{I}}, \boldsymbol{T}^b - \boldsymbol{T}_{select}^b), \quad (2)$$

where $\boldsymbol{T}_{select}^b$ denotes the subset of textual claims $\boldsymbol{T}^b$ associated with the evidence already selected.

*2) Juror executes fine-grained perception on the selected $\boldsymbol{e}_{*,t}^b$ to ground dispute-relevant pivotal clues $K_t^b$, along with the*

argument $A_t^b$ that supports or weakens the buyer's claims,

$$O_t^b : \{K_t^b, A_t^b\} = \mathcal{F}_{\mathrm{perceive}}(O_{\mathrm{I}}, \boldsymbol{e}_{*,t}^b, O_{t-1}^b). \quad (3)$$

Finally, the $\mathcal{F}_{\mathrm{select}}$ and $\mathcal{F}_{\mathrm{perceive}}$ process iterates until the buyer's claims are adequately verified, capped at $T_{max}$ rounds. The seller-side process is defined symmetrically. $O_{\mathrm{II}} : \{O_{T_{max}}^b, O_{T_{max}}^s\}$ provides the pivotal clues and reliable arguments of each disputant for subsequent stages.

▶ **Stage III: Adversarial Analysis.** To discern misleading claims from each disputant, the objective of IV-CoT shifts from *clue grounding* to *dispute causal analysis*. Upon acquiring pivotal clues, the juror uncovers the causal logic between the dispute focus and the adversarial clues, identifies logical contradictions $\Delta$, and summarizes the strengths and weaknesses of pivotal clues from each disputant into structured analysis reports $\Delta^b$ and $\Delta^s$,

$$O_{\mathrm{III}} : \{\Delta, \Delta^b, \Delta^s\} = \mathcal{F}_{\mathrm{analyze}}(O_{\mathrm{I}}, O_{\mathrm{II}}). \quad (4)$$

As illustrated in Fig. 4, the stage reveals a logical conflict: while the seller provides a video of the vacuum charging normally, the buyer offers immediate feedback upon receipt, highlighting a battery level of only 2%. In other words, the stage infers the underlying factors of the dispute, providing a profound logical basis for the final verdict.

▶ **Stage IV: Final Verdict.** Based on the previous outputs, juror $a_k$ evaluates the validity of both claims to reach a final verdict $\hat{y}_k$ and a traceable reason $J_k$,

$$O_{\mathrm{IV}} : \{\hat{y}_k, J_k\} = \mathcal{F}_{\mathrm{judge}}(O_{\mathrm{I}}, O_{\mathrm{II}}, O_{\mathrm{III}}). \quad (5)$$

In summary, IV-CoT achieves fine-grained pivotal clues grounding via a structured reasoning chain under limited context windows, while enhancing the clarity of the causal logic and the interpretability of verdicts.

## 4.2. Jury Consensus Verdict

Under flexible transaction rules, a single juror struggles to capture such dynamic characteristics, and is therefore more susceptible to inherent model biases (Taubenfeld et al., 2024). By contrast, collective discussion has been proven to improve reasoning quality and decision fairness (Chen et al., 2024; Arechar et al., 2023). To this end, we propose the Jury Consensus Verdict module, which incorporates a *Jury Simulation* mechanism, where multiple jurors with heterogeneous cognition exchange opinions, to mitigate individual biases. Furthermore, inspired by the principle of "*Stare Decisis*", JCV leverages verdict norms distilled from historical precedents to regulate current dispute decisions, thereby providing traceable guidelines for consensus verdict.

**Jury Simulation.** To mitigate individual cognitive biases, we design a jury simulation mechanism to emulate the interactive discussion of $N$ jurors over $T$ rounds. Specifically, given a dispute case $\boldsymbol{D}$, each juror $a_k$ participates in iterative interactions during round $t$, generating its decision $\hat{y}_{k,t}$ and reasoning $J_{k,t}$ through IV-CoT (Eq. (5)),

$$\hat{y}_{k,t}, J_{k,t} = \mathcal{F}_{\text{judge}}(\boldsymbol{D}, \boldsymbol{P}_k, \boldsymbol{M}_k, \boldsymbol{R}_{k,t}, \boldsymbol{S}_t). \qquad (6)$$

Here, each juror $a_k = \{\boldsymbol{P}_k, \boldsymbol{M}_k\}$ is composed of a unique persona $\boldsymbol{P}_k$, and an associated memory $\boldsymbol{M}_k$. $\boldsymbol{R}_{k,t}$ represents the verdict from jurors that $a_k$ follows in the network $\boldsymbol{G}$, and collective verdict summary $\boldsymbol{S}_t$ aggregates the decision reasons $J_{j,t-1}$ from all jurors in the round $t$-1,

$$\boldsymbol{R}_{k,t} = \{J_{j,t-1} \mid e_{k,j} \in \boldsymbol{E}\}, \boldsymbol{S}_t = \mathcal{F}_{\text{sum}}\left(d, \bigcup_{j=1}^{N} J_{j,t-1}\right). \qquad (7)$$

This design simulates the social dynamics of the jury $\boldsymbol{A}$, allowing each juror to maintain independent reasoning and update its verdict based on the logic of its social neighbors. Meanwhile, the summary $\boldsymbol{S}_t$ acts as a macro-level guideline, prompting outlier jurors to reconsider their verdicts, thereby enhancing the stability of collective decision.

**Verdict Precedent.** An important challenge when jurors make decisions in Eq. (6) is how to establish fair decisions. Inspired by the core principle of Stare Decisis in Common Law, we believe that EDV should also follow the verdict norms established in historical cases. Thus, we construct a Precedent Base $\boldsymbol{B} = \langle \boldsymbol{H}, \boldsymbol{N} \rangle$, where historical verdict records $\boldsymbol{H}$ are distilled into explicit verdict guidelines $\boldsymbol{N}$. To regulate the decision of juror $a_k$, we utilize the base $\boldsymbol{B}$ to personalize its memory, providing consistent and interpretable verdict grounds. Specifically, for a dispute $\boldsymbol{D}$, $\mathcal{F}_{\text{retrieve}}$ measures the semantic similarity between $\boldsymbol{D}$ and $\boldsymbol{H}$ to identify the most relevant precedent along with its corresponding guidelines $\boldsymbol{N}_{\text{guide}}$,

$$\boldsymbol{H}_{\text{guide}}, \boldsymbol{N}_{\text{guide}} = \mathcal{F}_{\text{retrieve}}(\boldsymbol{D}, \boldsymbol{B}). \qquad (8)$$

Here, $\boldsymbol{N}_{\text{guide}}$ contains several verdict norms $\boldsymbol{N}_j$ related to historical precedent $\boldsymbol{H}_{\text{guide}}$. We measure the similarity between these norms and the persona $\boldsymbol{P}_k$ with $\Phi(\cdot)$, selecting the top-$K$ norms as the juror's memory to guide its fair verdict,

$$\boldsymbol{M}_k = \{Rank(\Phi(\boldsymbol{P}_k, \boldsymbol{N}_j)) \leq K | \boldsymbol{N}_j \in \boldsymbol{N}_{\text{guide}}\} \qquad (9)$$

Further details about $\boldsymbol{B}$ are provided in Appendix B.2.

**Consensus Verdict.** As the rounds of simulation increase, the jury mechanism drives jurors' decisions toward a robust collective consensus. Finally, *CyberJurors* aggregates the votes for the seller and the buyer in the last round,

$$\hat{y} = \mathbb{I}\left(\hat{y}^s > \hat{y}^b\right), \hat{y}^b = \sum_{k=1}^{N} \mathbb{I}(\hat{y}_{k,T} = 0), \hat{y}^s = \sum_{k=1}^{N} \hat{y}_{k,T}. \quad (10)$$

The side $\hat{y}$ receiving more support votes wins the case, with the final collective verdict summary $\boldsymbol{S}_T$ serving as the verdict rationale. By integrating jury simulation with explicit precedent guidelines, *CyberJurors* significantly enhances the accuracy, fairness, interpretability, and social credibility of dispute verdict.

## 5. Experiments

**Configurations.** We conduct our experiments using the *Gemini-2.5-Flash-Lite-Nothinking* (Comanici et al., 2025). Experimental parameters are specified as follows, $T_{max} = 3$, $T = 3$, and $K = 3$. $\boldsymbol{G}$ is initialized according to the strategy proposed in (Mou et al., 2024; Sun et al., 2025; Zhou et al., 2026). To improve efficiency, we adopt an early stopping strategy when collective consensus exceeds a threshold $\delta = 0.8$. To optimize multimodal processing, we analyze video evidence via uniform sampling to mitigate information redundancy, with a strict maximum input limit of 30 frames per video.

**Baselines.** We evaluate the effectiveness of *CyberJurors* against five types of baselines: 1) Closed-Source LLMs include DeepSeek-V3 (Liu et al., 2024), Qwen-Plus, GPT-5.2-Chat, and GLM-4 (Zeng et al., 2024); 2) Open-Source LLMs include Dolphin3.0-R1-Mistral-24B and Llama-2-13B; 3) Closed-Source MLLMs include Gemini-3-Pro-Preview, Gemini-2.5-Flash-Lite-Nothinking, Doubao-Seed-1.6, Claude-Opus-4.5, GPT-5.2, and Grok-4; 4) Open-Source MLLMs include Llama-3.2-90B-Vision-Instruct and Qwen3-VL-235B-Instruct (Bai et al., 2025); 5) LLM-based Court Simulators include ChatEval (Chan et al., 2024) and AgentCourt (Chen et al., 2025a).

**Metrics.** To evaluate the efficacy of *CyberJurors* in dispute verdicts, we employ Accuracy (Acc.), Weighted F1 (Weig. F1), Macro F1 (F1), Macro Recall, and Macro Precision (Hossin & Sulaiman, 2015) to measure the alignment between predictions and ground-truth verdicts. Furthermore, Mean Absolute Error (MAE) and Root Mean Square Error (RMSE) are utilized to quantify the divergence between predicted support for disputants and actual jury voting.

Table 2. Performance Comparisons against Five Types Baselines. **Bold** and underline indicate the best and second-best results. The"↓"/"↑" indicates that smaller/higher values correspond to a closer match with the real-world results.

| Type | Method | Acc. ↑ | Weig. F1 ↑ | F1 ↑ | Recall ↑ | Precision ↑ | MAE ↓ | RMSE ↓ | Token ↓ |
|---|---|---|---|---|---|---|---|---|---|
| Closed LLM | DeepSeek-V3 | 0.6080 | 0.6042 | 0.6075 | 0.6593 | 0.6662 | - | - | 114,590 |
| | Qwen-Plus | 0.6055 | 0.5970 | 0.6037 | 0.6658 | 0.6826 | - | - | 146,381 |
| | GPT-5.2-Chat | 0.6344 | 0.6309 | 0.6340 | 0.6875 | 0.6954 | - | - | 158,189 |
| | GLM-4 | 0.5017 | 0.4655 | 0.4873 | 0.5823 | 0.6180 | - | - | 145,358 |
| Open LLM | Dolphin3.0-R1-24B | 0.4929 | 0.4568 | 0.4790 | 0.5747 | 0.6065 | - | - | 145,123 |
| | Llama-2-13B | 0.5042 | 0.4679 | 0.4898 | 0.5854 | 0.4224 | - | - | 144,899 |
| Closed MLLM | Gemini-3-Pro | 0.6354 | 0.6378 | 0.6351 | 0.6717 | 0.6670 | - | - | 4,370,185 |
| | Gemini-2.5-Flash-Lite | 0.5292 | 0.5166 | 0.5262 | 0.5876 | 0.5987 | - | - | 839,303 |
| | Doubao-Seed-1.6 | 0.5626 | 0.5567 | 0.5617 | 0.6139 | 0.6209 | - | - | 4,912,235 |
| | Claude-Opus-4.5 | 0.5910 | 0.5901 | 0.5910 | 0.6344 | 0.6357 | - | - | 2,698,359 |
| | GPT-5.2 | 0.4833 | 0.4907 | 0.4798 | 0.4944 | 0.4948 | - | - | 1,632,662 |
| | Grok-4 | 0.5436 | 0.5512 | 0.5363 | 0.5471 | 0.5441 | - | - | 1,188,798 |
| Open MLLM | Qwen3-VL-235B | 0.4843 | 0.4748 | 0.4825 | 0.5338 | 0.5367 | - | - | 2,988,145 |
| | Llama-3.2-90B-Vision | 0.4090 | 0.4187 | 0.3987 | 0.4018 | 0.4081 | - | - | 1,555,677 |
| Court Simulators | ChatEval | 0.6589 | 0.6645 | 0.6525 | 0.6678 | 0.6569 | - | - | 917,151 |
| | AgentCourt | 0.6673 | 0.6644 | 0.6383 | 0.6365 | 0.6413 | - | - | 75,280,342 |
| | Ours | **0.7292** | **0.7258** | **0.7037** | **0.6999** | **0.7100** | 4.7312 | 6.3724 | 62,327,703 |

## 5.1. Comparisons with State-of-the-Art

To evaluate the efficacy of *CyberJurors*, we conduct comprehensive comparative experiments on *VerdictBench*, as summarized in Table 2. Equipped with the IV-CoT and JCV modules, *CyberJurors* exhibits superior performance across all metrics. Specifically, it outperforms the second-best method with gains of 6.19% in Accuracy, 0.0613 in Weig. F1, 0.0512 in F1, 1.24% in Recall, and 1.46% in Precision. For an intuitive differences between *CyberJurors* and other baselines, we present some visualization results in Appendix C.2.

An intriguing empirical finding is that current MLLMs yield an average accuracy of 52.98%, which is notably inferior to the 55.78% achieved by text-only LLMs. This observation is consistent with our hypothesis that existing MLLMs, when processing ultra-long contexts, lack the capacity to fine-grainedly perceive pivotal visual clues, leading to degraded reasoning performance. In contrast, the superior performance of *CyberJurors* underscores the necessity of structured reasoning and the "select-perceive" iterative clues grounding stage for capturing granular multimodal details.

Furthermore, both *CyberJurors* and court simulators surpass the strongest single-model baseline *i.e.,* Gemini-3-Pro, with accuracy improvements of 9.38% and 3.19%, respectively. These gains are primarily attributed to the collective discussion, which enables jurors to reflect on their own decisions by referencing the decisions of neighboring jurors, effectively mitigating individual cognitive biases. However, a significant accuracy gap of 6.19% persists between *Cyber-Jurors* and the court simulators. This highlights our advantages in fine-grained content perception through structural

reasoning and explicit verdict guidelines derived from verdict precedents. Notably, *CyberJurors* achieves lower MAE and RMSE, demonstrating its superior ability to accurately align with the voting dynamics of the 17 platform jurors. In comparison, other baselines without 17-juror simulations cannot be evaluated on these alignment metrics.

## 5.2. Ablation Study

To validate the contribution of each component on EDV, we conduct a series of ablation experiments on validation *VerdictBench*, including our baseline, Rule Prompt, Structural Reasoning (SR-CoT), IV-CoT, Jury Simulation, and Verdict Precedents, as shown in Table 3.

Compared to our baseline, adding the verdict rule prompt yields a 4.60% accuracy improvement, demonstrating the positive contribution of the rule prompt detailed in Appendix E.7. Subsequently, SR-CoT (a four-stage reasoning process where Stage II is a one-step evidence selection) further boosts accuracy by 5.30% over the Rule Prompt. This reveals that domain-specific, structural logic reasoning is effective in parsing the intricacies of E-commerce disputes, confirming the necessity of a reasoning mechanism for complex decisions. Building upon this, the integration of IV-CoT, which introduces an "Select-Perceive" Iterative Clues Grounding strategy, further raises accuracy to 67.34%. This validates the importance of fine-grained perception of pivotal multimodal clues in EDV. Moreover, integrating the Jury Simulation pushes the accuracy to 70.18%, demonstrating the significant role of collective discussions and collective verdict summary in mitigating individual cognitive biases. Finally, *CyberJurors* incorporates explicit guidelines via Verdict Precedents, achieving the peak performance of

Table 3. Ablation Analysis on Each Component in *CyberJurors*.

| Method | Accuracy ↑ | Weig. F1 ↑ | Macro F1 ↑ | RMSE ↓ |
|---|---|---|---|---|
| Baseline | 0.5416 | 0.5433 | 0.5385 | - |
| + Rules | 0.5876 | 0.5887 | 0.5875 | - |
| + SR-CoT | 0.6406 | 0.6416 | 0.6810 | - |
| + IV-CoT | 0.6734 | 0.6788 | 0.6757 | - |
| + Jury | 0.7018 | 0.7043 | 0.6868 | **6.2106** |
| + Precedent | **0.7252** | **0.7196** | **0.6980** | 6.3234 |

Table 4. Performance and Token Comparisons.

| Method | Accuracy ↑ | Token ↓ |
|---|---|---|
| Gemini-2.5-Flash-Lite | 0.5500 | 1,037,520 |
| Gemini-3-Pro | 0.6354 | 4,120,022 |
| 1 juror with IV-CoT | 0.6734 | 2,862,538 |
| 1 juror with IV-CoT-Lite | 0.6600 | 1,881,823 |
| 17 jurors within JCV | 0.7252 | 49,746,582 |

72.52%. In summary, integrating IV-CoT and JCV modules, *CyberJurors* significantly improves the decision consistency and verdict fairness, while better aligning with the voting dynamics of real-world 17 jurors.

### 5.3. Token Consumption

To quantitatively evaluate the computational overhead of dispute verdict, we compare the token consumption of different methods on 100 randomly selected cases, as reported in Table 2. In general, MLLM-based methods consume more tokens than text-only LLMs, with the highest token usage among MLLMs reaching 4.9122M. Additionally, court simulators are substantially more expensive than single-model approaches: ChatEval consumes 0.9172M tokens, exceeding all LLM baselines, while AgentCourt requires 75.2803M tokens, surpassing every MLLM baseline. Despite being a multimodal court simulator, *CyberJurors* consumes 62.3277M tokens, lower than AgentCourt, while still attaining the highest accuracy. This efficiency advantage is mainly attributed to the fine-grained multimodal perception mechanism of IV-CoT.

To ensure a fairer comparison, we also compare the performance of IV-CoT against strong baselines under equivalent compute budgets, as shown in Table 4. Compared to Gemini-3-Pro, IV-CoT achieves a 3.80% improvement in Accuracy while reducing token consumption by 30.5%. This stems from IV-CoT's structured decomposition of the EDV task, which directly targets the identified bottlenecks of existing approaches rather than merely relying on a larger token budget. Furthermore, a lightweight variant, IV-CoT-Lite (which retains our core innovations while streamlining secondary inputs), consumes about 1.8× the tokens of Gemini-2.5-Flash-Lite but achieves a significant 11.0% improvement in accuracy. Meanwhile, extending IV-CoT to 17 jurors within JCV further raises the accuracy to 72.52%. However, JCV is

not designed solely to mitigate individual bias or improve accuracy. Importantly, it aims to simulate the "crowdsourced jury" mechanism in E-commerce platforms and to replicate the voting distribution of the 17 human jurors. JCV should be viewed as a necessary architectural choice for modeling real-world verdict dynamics rather than a simple alternative to a single-model setup.

### 5.4. Performance Analysis

To assess the effectiveness of *CyberJurors*, we analyze the jury discussion dynamics, and benchmark the framework against our baseline and AgentCourt across varying difficulty levels and product categories.

**Jury Discussion.** As illustrated in Fig. 5(a), we analyze the distribution of discussion rounds across varying difficulty levels, together with the corresponding verdict accuracy. *CyberJurors* can adaptively adjust the number of discussion rounds according to dispute complexity, thereby improving verdict efficiency. For low-difficulty cases, JCV typically triggers early stopping after Round 1, with consistent verdict accuracy exceeding 80%. As difficulty increases, the proportion of cases requiring more rounds rises markedly, mirroring a more cautious human-like decision process. Notably, the effect of "Collective Consensus" is particularly evident in buyer-win cases, consistent with the ablation findings in Table 3. For example, in disputes with 11:6 ground-truth votes, the independent 17-juror verdicts in Round 1 attain only 36.84% accuracy, whereas introducing JCV can mitigate individual biases, thereby yielding a substantial 9.21% improvement.

**Difficulty Levels.** As shown in Fig. 5(b), *CyberJurors* achieves the best performance across difficulty levels, demonstrating consistent advantages in accuracy, robustness, and decision consistency. Such results stem from the integration of IV-CoT, particularly its "Select-Perceive" Iterative Clues Grounding strategy, with JCV guided by Verdict Precedents. Specifically, for low-difficulty cases where the reasoning path is relatively direct, *CyberJurors* attains optimal results in Round 1 (Fig. 5(a)), indicating that IV-CoT efficiently aligns the dispute focus with pivotal clues. Conversely, other methods lag behind in these scenarios due to limited ability to perceive fine-grained visual clues. For high-difficulty cases, both *CyberJurors* and AgentCourt significantly outperform our baseline through collective discussion, while *CyberJurors* yields more reliable gains than AgentCourt with the help of Verdict Precedents.

**Product Category.** Figure 5(c) reveals that *CyberJurors* performs consistently well across all product categories, with particularly notable gains in *Digital & Appliances* and *Home & Lifestyle*. Referring to Fig. 3(d), these two categories are inherently more difficult to make verdicts. The former often involves multi-dimensional disputes (*e.g.*, physical damage,

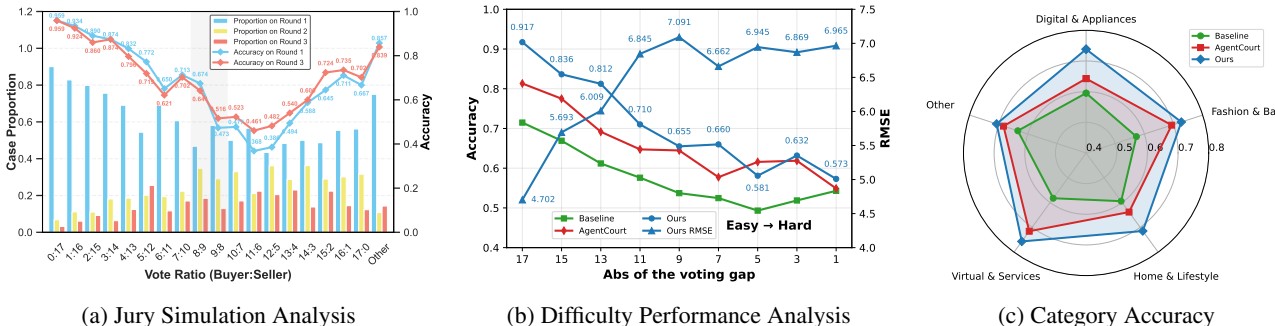

(a) Jury Simulation Analysis         (b) Difficulty Performance Analysis         (c) Category Accuracy

Figure 5. Performance Analysis of *CyberJurors*. (a) investigates the case proportion with different rounds and the accuracy under varying difficulty. (b-c) benchmark the robustness of *CyberJurors* against the baseline and AgentCourt across varying difficulty and categories.

functional defects, and authenticity verification), with more complex evidence chains, posing higher demands on fine-grained multimodal evidence perception. The latter typically centers on appearance, size, and usage traces, which rely more heavily on transaction rules. Therefore, the superior performance in these high-complexity categories can be attributed to our unique design: IV-CoT enables fine-grained clues extraction and dispute-focused causal reasoning, while JCV improves verdict reliability through collective consensus and explicit guideline integration.

### 5.5. Case Study

To more intuitively demonstrate the advantages of *Cyber-Jurors*, we present an in-depth case study in Fig. 6. The central dispute is whether the hat exhibits undisclosed usage traces or whether such traces are fully disclosed by the seller before purchase. Results reveal that existing methods fail to deliver a correct verdict. Specifically, *GPT* cannot perceive multimodal evidence and reaches the erroneous conclusion that the seller "failed to disclose" with individual bias. While *Gemini* identifies visual clues "noticeable yellowing", it produces a hallucinated claim "97% new", which further results in misjudgment. Although *AgentCourt* identifies the intention of each disputant through adversarial reasoning, its judge, lacking explicit guidelines, tends to favor the buyer, leading to an incorrect verdict. Conversely, our 17 jurors accurately capture pivotal visual evidence aligned with the dispute focus, and deliver diverse, specialized perspectives. Ultimately, by applying explicit guidelines, *CyberJurors* corrects individual biases and successfully replicates the ground-truth voting distribution. These findings indicate that *CyberJurors* meets the two core needs of EDV: clear clue-dispute causal logic and persuasive verdicts.

### 6. Conclusions

We introduce a pioneering task, E-commerce Dispute Verdict, and construct *VerdictBench*, the first multimodal dispute benchmark, designed to support fair and intelligent dispute verdicts. Its inherent complexity paves new avenues

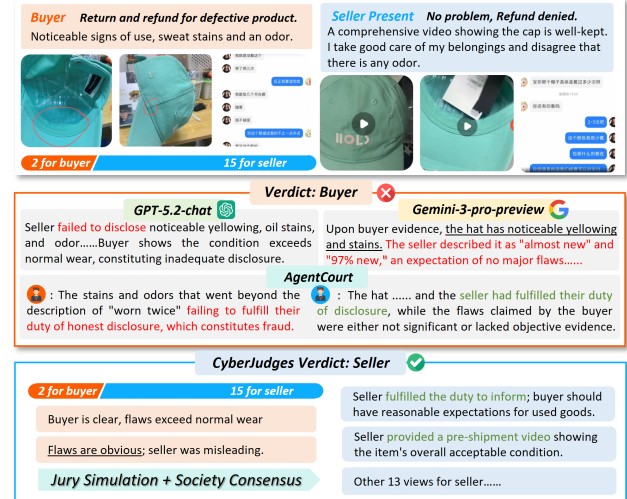

Figure 6. Visualization of a Case Study. Red text highlights the reasoning errors, underlined areas indicate the focused multimodal regions, and green text marks pivotal clues for the verdict.

for research in intent reasoning and multi-agent simulation. Moreover, we propose *CyberJurors*, which integrates Individual Verdict Chain-of-Thought with Jury Consensus Verdict, faithfully simulating jury-style collective decision-making. *CyberJurors* outperforms existing LLMs, MLLMs, and court simulators by a substantial margin, demonstrating superior ability to capture pivotal clues, mitigate individual biases, and enhance decision interpretability.

Looking ahead, we will explore fine-tuning strategies to better align *CyberJurors* with domain-specific verdict regulations. We also intend to advance fine-grained evidence perception capabilities. Furthermore, we will explore the Generative Social Choice framework (Boehmer et al., 2025) to move beyond binary verdicts by generating compromise mediation plans, thereby ensuring that diverse and reasonable views are fairly represented in the final decision. These efforts are directed toward the ultimate goal of achieving robust, fair, and intelligent E-commerce dispute verdicts, eventually providing a highly efficient alternative to time-consuming "crowdsourced jury" mechanism.

## Acknowledgements

This work was supported by the National Key Research and Development Program of China (NO. 2024YFE0203200), the National Nature Science Foundation of China (NO. U24A20329, NO. 62527810), the Fundamental and Interdisciplinary Disciplines Breakthrough Plan of the Ministry of Education of China (NO. JYB2025XDXM103), and the Science Fund for Creative Research Groups (No.62121002).

## Impact Statement

The development and application of the E-commerce dispute verdict systems require careful ethical consideration. First, the data collection process for *VerdictBench* has been approved by our Academic Committee. We have implemented a strict de-identification mechanism to protect users' sensitive information, including names, contact details, and addresses. Only content relevant to the disputes is retained, thereby eliminating risks of privacy infringement. Second, *CyberJurors* is intended to support robust, fair, and intelligent dispute verdicts in E-commerce scenarios, serving as an equitable alternative to time-consuming human juries. In such contexts, transparency and strict governance are essential to mitigate potential misuse in real-world dispute verdicts. This includes clearly defining the scope of use, enforcing access control, and establishing ethical boundaries to avoid unintended societal harm. We commit to adhering to these principles and to advancing the research in a lawful and compliant manner, ensuring that *CyberJurors* contributes to the responsible development of E-commerce dispute verdicts.

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

In this appendix, we provide additional details on dataset construction and statistical analysis (Sec. A), method details including the algorithm of *CyberJurors* and the construction process of the Verdict Precedent Base (Sec. B), qualitative results including the experimental setting, visual comparisons with state-of-the-art methods, and stability analysis (Sec. C), as well as limitations (Sec. D) and the complete prompt set used in *CyberJurors* (Sec. E).

## A. More Dataset Details

This section provides supplementary information on *VerdictBench*, detailing the dataset construction process and statistical analysis.

### A.1. Dataset Construction

To construct a high-quality, interpretable, and ethically compliant E-commerce dispute dataset, we design a standardized data construction pipeline, as illustrated in Fig. 2.

**Data Collection.** E-commerce platforms employ a random assignment mechanism, where each case is evaluated by 17 anonymous jurors who provide votes and rationales separately. Once all 17 votes for a dispute have been submitted, the platform gives a final verdict based on the voting results and executes the corresponding financial processing. After obtaining approval from the Academic Committee, we recruit 14 volunteers with higher education backgrounds and collect their historical dispute cases within the authorized scope. The data we collected fully retains the multi-round textual claims and multimodal evidence from the actual process, thereby supporting the evaluation of verdict prediction and evidence reasoning capabilities.

**Quality Checker.** To ensure the usability of subsequent structured processing and experimental evaluation, we first conduct quality validation. This procedure comprises three stages—*Structural Consistency*, *Semantic Alignment*, and *Metadata Tagging*—aimed at removing noise, increasing the informational density of the corpus, and mitigating experimental risks arising from structural omissions or unreadable media.

- **Structural Consistency**: We examine the structural consistency of the raw files to ensure that critical fields are present and can be parsed reliably. For missing fields, we attempt re-collection or manual completion; if the required information remains unobtainable, the corresponding case is discarded.

- **Semantic Alignment**: We verify whether media indices referenced in the text correspond to local media files, and assess the readability and validity of those files. Missing or corrupted media are re-collected when possible; if recovery fails, the corresponding case is discarded.

- **Metadata Tagging**: We assign category tags to support statistical analysis. Specifically, we use *GPT- 4o-mini* to select the most appropriate label for each product from a predefined category set (five top-level categories and their associated subcategories), as illustrated in Fig. 3 (a). Using the prompt in Appendix E.10, we invoke *GPT- 4o-mini* on each case's `product_text` to perform tagging, thereby producing category tags suitable for stratified sampling.

**Protect Privacy.** After completing quality validation, we de-identify each case and apply redaction to images when necessary. Building upon the platform's existing privacy-protection mechanisms, volunteers flag cases during the review process that may contain sensitive information; based on these flags, we conduct manual verification and perform de-identification on these content. Cases that cannot be remediated within compliance constraints are removed to ensure that the dataset is suitable for public research and reproducible experimentation.

**Case Formatting.** To improve readability and enable downstream modeling to access conversational rounds, roles, and evidence types with precision, we use regular-expression matching to convert each case into a JSON format that is consistent with the original structure. We then perform merging and deduplication to remove duplicate records and redundant samples. In addition, to characterize the overall stability of volunteers' participation in the review process, we analyze each volunteer's review accuracy using the platform's final verdict as the reference. The results are reported in Table 5: accuracy varies across volunteers, indicating that individual differences affect final verdicts; the overall mean accuracy is 80.3%, which also suggests that the task is highly contentious and challenging in real-world settings. Finally, over the continuous collection and processing period from May 13, 2025 to December 22, 2025, we filter 6,000 high-quality cases from 8,471 candidate records and construct *VerdictBench*.

Table 5. Volunteer-level Verdict Accuracy. It shows the number of reviewed cases and the corresponding accuracy against the final verdict for each recruited volunteer.

| Volunteer_id | 1 | 2 | 3 | 4 | 5 | 6 | 7 | 8 | 9 | 10 | 11 | 12 | 13 | 14 | Total |
|---|---|---|---|---|---|---|---|---|---|---|---|---|---|---|---|
| Case Count | 698 | 503 | 860 | 716 | 747 | 329 | 547 | 306 | 665 | 892 | 738 | 573 | 568 | 329 | 8471 |
| Acc(%) | 82.2 | 78.2 | 77.5 | 78.4 | 76.8 | 83.5 | 90.9 | 81.9 | 84.2 | 79.8 | 77.1 | 78.6 | 75.7 | 79.5 | 80.3 |

**Data Partitioning.** To make performance evaluation more reliable, we first examine the statistical features of the dataset before partitioning it, as shown in Fig. 3. Since both the product category and the case difficulty influence the final verdicts, we partition the data using a joint partitioning approach based on "category–difficulty level" to create the training, validation, and test sets in a ratio of 3:1:2. Difficulty is measured by the vote split between the buyer and the seller among 17 jurors. As shown in Table 6, after partitioning, each subset maintains category and verdict label distributions that match the overall *VerdictBench*.

Table 6. Statistics of the Data Partitioning in *VerdictBench*. It shows the numbers of cases in the training, validation, and test sets across five coarse-grained product categories and final verdict labels.

| | Counts | Digital | Fashion | Home | Virtual | Other | Buyer Wins | Seller Wins |
|---|---|---|---|---|---|---|---|---|
| *Train* | 3009 | 773 | 705 | 474 | 506 | 551 | 1128 | 1881 |
| *Val* | 986 | 256 | 232 | 152 | 166 | 180 | 367 | 619 |
| *Test* | 2005 | 516 | 472 | 319 | 333 | 365 | 747 | 1258 |
| *Total* | 6000 | 1545 | 1409 | 945 | 1005 | 1096 | 2242 | 3758 |

## A.2. Dataset Statistics

To systematically characterize the real-world distributional properties and task challenges of *VerdictBench*, we conduct a statistical analysis from multiple perspectives, as shown in Fig. 3. These include category coverage, verdict distribution, verdict difficulty, and the scale of multimodal evidence. To facilitate reproducibility and verification, we further specify the plotting procedure for each subfigure, the meaning of the axes, and the primary empirical observations in Table 7.

Table 7. Detailed descriptions of Fig. 3. It includes the plotting protocol, axis definitions, and main empirical observations for each subfigure.

| Stat Name | Description | Conclusions |
|---|---|---|
| Category Distribution Fig. 3 (a) | Visualizes the dataset's taxonomic structure. Samples are classified into a two-level hierarchy with five top-level categories and 25 subcategories; we use *GPT-4o-mini* to ensure semantic consistency. The donut chart shows the proportional composition of each top-level category. | The dataset achieves comprehensive scenario coverage and spans all 25 defined subcategories. The top-level distribution is relatively balanced, with each category exceeding 15%. This mitigates class-imbalance bias and supports robust evaluation across diverse E-commerce domains. |
| Overall Win Rate Fig. 3 (b) | Shows the overall distribution of verdict outcomes based on the human jury. The chart contrasts the win rates of the two disputants across the entire dataset. | Sellers exhibit a distinct advantage, winning 62.6% of cases, compared with 37.4% for buyers. This likely reflects sellers' greater familiarity with platform policies and more standardized evidence retention, which together create a systematic advantage in verdicts. |
| Category Win Rate Fig. 3 (c) | Decomposes win rates by the five top-level product categories to reveal category-specific verdict tendencies. Bars show the normalized proportion of buyer wins and seller wins within each category. | Win rates vary substantially across domains. *Virtual & Services* strongly favors sellers, with a 75.9% seller win rate. *Digital & Appliances* is more competitive, with a 55.9% seller win rate. This suggests that disputes over tangible goods are more contestable than those involving services. |

Table 7. Detailed descriptions for Fig. 3 (continued).

| Stat Name | Description | Conclusions |
|---|---|---|
| Difficulty Distribution Fig. 3 (d) | Measures case difficulty using vote margins from 17 human jurors. We group outcomes by the absolute split $\lvert b - s \rvert$ and merge symmetric patterns $b : s$ with $s : b$, ordering results from unanimous decisions 17:0 (easiest) to near ties 9:8 (hardest). Bars show the overall percentage at each split, stacked by buyer and seller advantage. Overlaid curves show within-category percentages for each top-level category, enabling direct comparison of difficulty profiles across categories. | The share of cases decreases as difficulty increases, consistent with the real-world pattern that most disputes are straightforward, while only a minority are genuinely hard. Difficulty profiles also differ systematically across categories. *Digital & Appliances* and *Home & Lifestyle* skew toward harder cases: their unanimous-consensus rates fall below the overall 13.9%, while their most-contentious rates exceed the overall 9.1%. The former often involves multi-dimensional disputes such as physical damage, functional defects, and authenticity verification, leading to longer evidentiary chains and higher demands for fine-grained multimodal understanding. The latter more frequently centers on appearance, size, and signs of use, and tends to be more tightly constrained by transaction rules. By contrast, *Virtual & Services* is relatively easier, plausibly because platform rules often default to non-returnability for virtual goods. |
| Media Distribution Fig. 3 (e) | Analyzes multimodal density by counting distinct media assets per case. The chart separates image counts on the upper axis and video counts on the lower axis, and uses dashed lines to mark population means. | *VerdictBench* is highly multimodal, averaging 14.1 images and 0.9 videos per case. The modal image count lies in the 5–10 range, covering 1,501 cases. In addition, 2,083 cases include video evidence, underscoring the need for robust visual processing. |
| Evidence Intensity Fig. 3 (f) | Examines the number of evidence pieces submitted by each disputant role. The bi-directional bar chart compares the frequency distribution of evidence counts for buyers on the upper axis versus sellers on the lower axis. | Evidence intensity is markedly asymmetric. Buyers submit substantially more evidence, with a mean of 5.5, while sellers average 2.5. This suggests buyers often adopt an exhaustive accumulation strategy, whereas sellers tend to provide more targeted and decisive counter-evidence. |
| Evidence Analysis Fig. 3 (g) | Examines whether the *relative* intensity of buyer versus seller evidence is associated with the final verdict. We construct three evidence-intensity gap metrics at the case level: (i) **Evidence Count Gap**, defined as the difference between the number of buyer and seller evidence submissions; (ii) **Text Length Gap**, defined as the difference between the total character length of buyer and seller evidence texts (summing over all submissions per side); and (iii) **Media Count Gap**, defined as the difference between the total number of attached media files (images and videos) in buyer and seller submissions. For each metric, we compute a normalized gap score $$\frac{\text{Buyers}' - \text{Sellers}'}{\text{Buyers}' + \text{Sellers}'}.$$ The x-axis shows the corresponding normalized gap, where positive values indicate a buyer advantage and negative values indicate a seller advantage; the y-axis enumerates the three metrics. Each point represents one case, with jitter applied along the y-direction for readability. Point color indicates the verdict outcome, and marker shape encodes the metric | The distribution of points exhibits no visually discernible clustering or monotonic trend across any of the three metrics. Overall, these results suggest that simple quantitative proxies of evidence intensity (more submissions, longer text, or more media attachments) are not reliably predictive of verdict outcomes. |

# B. More Method Details

## B.1. Algorithm

The overall computational workflow of the proposed framework *CyberJurors* is formally presented in **Algorithm 1**.

---

**Algorithm 1** *CyberJurors*: A Multi-Agent Simulation Task for E-Commerce Disputes Verdict

---

1: **Input:** Case Data $D = \{d, e_1^b, e_1^s, \cdots, e_{N_b}^b, e_{N_s}^s\}$, Juror Set $A = \{a_1, \ldots, a_{17}\}$, Jury Discussion Rounds $T$, Precedent Base $B = \langle H, N \rangle$, Clues Grounding Rounds $T_{max}$, and threshold $\delta$.
2: **Output:** Verdict result $\hat{y}$ and collective verdict summary $S_T$.
3: Generate social network $G = \langle A, E \rangle$, and initialize each juror $a_k$ with $P_k$;
4: Initialize collective verdict summary $S_0 \leftarrow \emptyset$;
5: **for** round $t = 1$ **to** $T$ **do**
6:    **for each** juror $a_k \in A$ **do**
7:       $O_{\mathrm{I}} \leftarrow \mathcal{F}_{\mathrm{extract}}(d, T^b, T^s)$ to extract dispute focus and claims from text evidence and transaction information;
8:       **for** iteration $t_b = 1$ **to** $T_{max}$ **for** buyer $b$ **do**
9:          **Evidence Selection:** Select pivotal evidence $e_{*,t_b}^b = \mathcal{F}_{\mathrm{select}}(O_{\mathrm{I}}, T^b - T_{select}^b)$;
10:          **Fine-grained Perception:** $O_{t_b}^b \leftarrow \{K_{t_b}^b, A_{t_b}^b\} = \mathcal{F}_{\mathrm{perceive}}(O_{\mathrm{I}}, e_{*,t_b}^b, O_{t_b-1}^b)$ to extract dispute-relevant clues and arguments;
11:          **if** visual evidence is sufficient **then**
12:            **break**;
13:          **end if**
14:       **end for**
15:       **for** iteration $t_s = 1$ **to** $T_{max}$ **for** seller $s$ **do**
16:          **Evidence Selection:** Select pivotal evidence $e_{*,t_s}^s = \mathcal{F}_{\mathrm{select}}(O_{\mathrm{I}}, T^s - T_{select}^s)$;
17:          **Fine-grained Perception:** $O_{t_s}^s \leftarrow \{K_{t_s}^s, A_{t_s}^s\} = \mathcal{F}_{\mathrm{perceive}}(O_{\mathrm{I}}, e_{*,t_s}^s, O_{t_s-1}^s)$ to extract dispute-relevant clues and arguments;
18:          **if** visual evidence is sufficient **then**
19:            **break**;
20:          **end if**
21:       **end for**
22:       $O_{\mathrm{II}} \leftarrow \{O_{T_{max}}^b, O_{T_{max}}^s\}$ by aggregating all collected visual clues and arguments ;
23:       $O_{\mathrm{III}} \leftarrow \mathcal{F}_{\mathrm{analyze}}(O_{\mathrm{I}}, O_{\mathrm{II}})$ to detect contradictions $\Delta$ from the selected clues of both disputants and obtain structured analysis reports $\Delta^b$ and $\Delta^s$;
24:       Receive verdict outcomes from neighbors $R_{k,t} \leftarrow \{J_{j,t-1} \mid e_{k,j} \in E\}$ from social network $G$;
25:       Retrieve the most relevant precedents with their guidelines $H_{\mathrm{guide}}, N_{\mathrm{guide}} \leftarrow \mathcal{F}_{\mathrm{retrieve}}(D, B)$;
26:       Select $K$ norms as the juror's memory $M_k \leftarrow \{Rank(\Phi(P_k, N_j)) \leq K | N_j \in N_{\mathrm{guide}}\}$;
27:       Generate decision and reasoning $\{\hat{y}_{k,t}, J_{k,t}\} \leftarrow \mathcal{F}_{\mathrm{judge}}(O_{\mathrm{I}}, O_{\mathrm{II}}, O_{\mathrm{III}}, P_k, M_k, R_{k,t}, S_t)$;
28:    **end for**
29:    Update collective verdict summary $S_{t+1} \leftarrow \mathcal{F}_{\mathrm{sum}}(d, \bigcup_{j=1}^N \{J_{j,t}\})$;
30:    Aggregate votes for both disputants $\hat{y}_t^b = \sum_{k=1}^{17} \mathbb{I}(\hat{y}_{k,t} = 0)$ and $\hat{y}_t^s = \sum_{k=1}^{17} \hat{y}_{k,t}$;
31:    Calculate collective consensus $c \leftarrow \frac{1}{17} \max(\hat{y}_t^b, \hat{y}_t^s)$
32:    **if** $c > \delta$ **then**
33:       **break**;
34:    **end if**
35: **end for**
36: $\hat{y}^b = \sum_{k=1}^{17} \mathbb{I}(\hat{y}_{k,T} = 0), \hat{y}^s = \sum_{k=1}^{17} \hat{y}_{k,T}$;
37: Determine final winner by majority vote at termination $\hat{y} = \mathbb{I}(\hat{y}^s > \hat{y}^b)$;
38: **return** Final verdict $\hat{y}$ and collective verdict summary $S_T$.

---

## B.2. Verdict Precedent Construction

The construction of the Precedent Base $B = \langle H, N \rangle$ aims to transform real verdict records $H$ into explicit and retrievable verdict guidelines $N$ the reasoning and perception capabilities of MLLMs. These verdict records $H$ are sourced from the training set of *VerdictBench*. The base construction process consists of two core stages.

Initially, we use an LLM to perform a deep analysis of the standardized textual information of a precedent $h_i \in H$. It distills a universal verdict guideline $n$ from the real decisions and rationales of the 17 human jurors $\{j_{i,k}\}_{k=1}^{17}$:

$$n_i = \mathcal{F}_{\mathrm{reflect}}(d_i, T_i^b, T_i^s, \{j_{i,k}\}_{k=1}^{17}). \tag{11}$$

Here, the generated guideline $n$ consists of multiple verdict norms, and $\mathcal{F}_{\text{reflect}}$ represents the function for guideline reflection prompt as detailed in Appendix E.9. To achieve efficient retrieval, a pre-trained model `Longformer-base-4096` is used as the semantic encoder $\mathcal{F}_{\text{encoder}}$ to map each precedent $h_i$ into a high-dimensional feature vector $v_i = \mathcal{F}_{\text{encoder}}(h_i)$, thus establishing a semantic association index.

Next, to ensure the consistency of the Precedent Base across diverse cases, we incorporate a dynamic closed-loop construction mechanism. During base expansion, when a new precedent $h_{new}$ is introduced, we calculate the cosine similarity between its feature vector $v_{new}$ and the historical feature vectors $v_i$ already in the base:

$$\text{sim}(v_{new}, v_i) = \frac{v_{new} \cdot v_i}{\|v_{new}\|\|v_i\|} = \frac{\sum_{j=1}^{m} v_{new,j} v_{i,j}}{\sqrt{\sum_{j=1}^{m} v_{new,j}^2} \sqrt{\sum_{j=1}^{m} v_{i,j}^2}}. \tag{12}$$

We retrieve the most relevant historical guidelines $N_{\text{guide}} = \mathcal{F}_{\text{retrieve}}(h_{new}, B)$. Based on these, the reflection guideline $n_{new}$ for the new case is refined and generated:

$$n_{new} = \mathcal{F}_{\text{reflect}}(d_{new}, T_{new}^b, T_{new}^s, \{j_{new,k}\}_{k=1}^{17}, N_{\text{guide}}). \tag{13}$$

Subsequently, $n_{new}$ is dynamically incorporated into the set of verdict guidelines $N$.

This dynamic mechanism ensures that new guidelines are no longer isolated logical verdicts but are rooted in the evolutionary results of recognized verdict norms. Through this process, the Precedent Base achieves spontaneous evolution, significantly enhancing the consistency and interpretability of verdict standards. Ultimately, this ensures that the verdict logic deeply aligns with human notions of fairness and social values.

## C. More Qualitative Results

### C.1. Experimental Setting

To ensure the reproducibility and transparency of our results, all experiments are executed on a server with a 32-vCPU Intel(R) Xeon(R) Platinum 8352V CPU @ 2.10GHz and 240GB of RAM. For all generative models, the temperature is uniformly set to 0.7 to balance creativity and consistency, with a maximum output constraint of 2,048 tokens. We use Gemini-2.5-Flash-Lite-Nothinking as the core engine for court simulation because it provides a strong balance between multimodal understanding and reasoning efficiency. Furthermore, to clarify the handling of multimodal evidence within *VerdictBench*, we specify that for all text-only LLM baselines, visual evidence is entirely omitted. Similarly, for the LLM-based court simulators, we maintain their original text-only architectural frameworks without additional adaptation for multimodal input. As a result, visual information is also ignored during simulation.

### C.2. Visual Comparisons with State-of-the-Art Methods

To conduct a fine-grained analysis of the experimental results, we statistically measure the accuracy of each method across different ground-truth vote ratios. Fig. 7(a) compares the accuracy of *CyberJurors* with that of various generative models. The study reveals that existing generative models suffer from severe bias: their accuracy on cases that are actually won by the seller is significantly lower than on cases won by the buyer, showing a systematic tendency to predict a buyer victory. Taking *DeepSeek-V3* as an example, the accuracy gap between cases with 8:9 and 9:8 ground-truth votes reaches 66.5%. By contrast, *CyberJurors* exhibits a much smaller accuracy difference between cases with different winners, underscoring the importance of Jury Simulation and verdict precedents for bias mitigation.

Furthermore, we present the accuracy of *CyberJurors* under different predicted vote margins in Fig. 7(b). The figure shows that the predicted vote margins tend to be lopsided, indicating higher internal consistency of the model and demonstrating the cost-saving value of the early-stopping strategy. Moreover, accuracy steadily rises as the vote gap widens, increasing by 20.6% from cases predicted as 3:14 to those predicted as 0:17. Notably, *CyberJurors* achieves 87.9% accuracy when predicting 0:17, surpassing the average performance of human jurors (Table 5) and marking an important step toward intelligent E-commerce dispute verdicts. This highlights the real-world deployment potential of *CyberJurors*, and future work will focus on improving accuracy across other predicted vote outcomes.

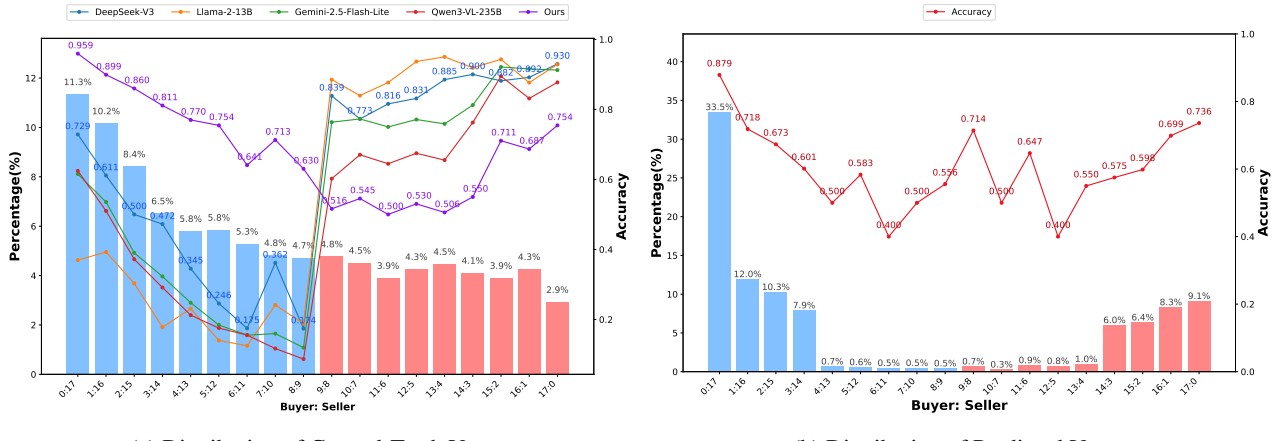

(a) Distribution of Ground-Truth Votes.

(b) Distribution of Predicted Votes.

Figure 7. Visual Comparisons of Votes Distribution.

Table 8. Performance and stability comparison across five categories of baselines. Accuracy measures predictive performance, while the $P$-value from Cochran's Q test reflects run-to-run stability.

| Type | Method | Acc. ↑ | $P$-value ↑ |
|---|---|---|---|
| Closed LLM | DeepSeek-V3 | 0.6080 | 0.7008 |
| | Qwen-Plus | 0.6055 | 0.6116 |
| | GPT-5.2-Chat | 0.6344 | 0.6834 |
| | GLM-4 | 0.5017 | 0.3820 |
| Open LLM | Dolphin3.0-R1-24B | 0.4929 | 0.7469 |
| | Llama-2-13B | 0.5042 | 0.7675 |
| Closed MLLM | Gemini-3-Pro | 0.6354 | 0.1114 |
| | Gemini-2.5-Flash-Lite | 0.5292 | 0.5705 |
| | Doubao-Seed-1.6 | 0.5626 | 0.3159 |
| | Claude-Opus-4.5 | 0.5910 | 0.7623 |
| | GPT-5.2 | 0.4833 | 0.2311 |
| | Grok-4 | 0.5436 | 0.5740 |
| Open MLLM | Qwen3-VL-235B | 0.4843 | 0.1331 |
| | Llama-3.2-90B-Vision | 0.4090 | - |
| Court Simulators | ChatEval | 0.6589 | 0.7210 |
| | AgentCourt | 0.6673 | 0.7051 |
| | Ours | **0.7292** | 0.9098 |

### C.3. Stability Analysis

To ensure experimental rigor, we conduct a stability analysis experiment for every comparative method, as reported in Table 8. We randomly sample the same 100 cases and run 5 independent repetitions on each method, followed by Cochran's Q test (Borenstein et al., 2021) on the results.

According to the P-value listed in Table 8, all methods are stable. Moreover, court simulators are more stable than any single model: the least stable simulator, AgentCourt, achieves $P = 0.7051$, and only three single models exceed this value. This demonstrates the robustness of jury simulation. *CyberJurors* further attains the highest P-value of 0.9098, illustrating that the multi-round simulation and verdict precedents can mitigate incidental biases.

Moreover, the same Q test analysis is applied to each setting of the ablation experiments. As shown in Table 9, based on the $P$-value, we fail to reject the null hypothesis, indicating that the five independent runs exhibit no statistically significant differences. Further, to verify the significance of the incremental gains, we conduct a paired-sample $T$-test between adjacent modules. Under the chosen significance level, all calculated $T$-statistics are larger than the critical value of 2.132, showing that the module-level improvements are statistically significant. The two results prove that the incremental improvements of *CyberJurors* are statistically significant, highly stable, and not artifacts of variance.

Table 9. Statistical Significance Analysis of Components.

| Method | $P$-value | $T$-statistic |
|---|---|---|
| baseline | 0.5705 | – |
| SR-CoT | 0.7832 | 11.0297 |
| IV-CoT | 0.7640 | 4.3637 |
| Jury | 0.8392 | 6.3500 |
| Precedent | 0.9098 | 3.6770 |

## D. Limitations

While *CyberJurors* has demonstrated strong potential for E-commerce dispute verdicts, several key areas merit further exploration before fully automated and impartial intelligent verdict can be achieved.

First, existing generative language models often exhibit significant individual cognitive biases when handling transaction disputes. As illustrated in Fig. 7(a) of Appendix C.2, their accuracy in cases where the seller actually wins is substantially lower than in those where the buyer wins. This "pro-consumer" bias is particularly prominent when processing ambiguous dispute focus. To mitigate such biases, our future work will focus on Supervised Fine-Tuning (SFT), using the jurors' rationales in *VerdictBench* as alignment signals to guide models toward fairer behavior in the E-commerce domain.

Second, in real-world scenarios, jurors originate from diverse social backgrounds, and their verdicts are deeply influenced by personal cognitive preferences and professional expertise. Therefore, future research will be dedicated to constructing a heterogeneous jury composed of multiple models. By introducing a variety of MLLMs with different parameter scales and inductive biases, together with richer personas and personality traits, we aim to build a dispute verdict simulator that more closely reflects real-world social decision-making processes.

Finally, the size of the jury is a critical variable affecting decision quality and computational cost. Although we adopt a standard scale of 17 jurors, whether an optimal jury size exists remains an open question. In the future, we will investigate dynamic jury mechanisms that adaptively adjust the number of participating jurors based on the complexity of the target case.

## E. Prompt Set

### E.1. Here, we provide the detailed prompts used in our baselines:

---

**Prompt E.1: Prompt for baselines of MLLMs & LLMs**

You are a professional juror for disputes on a second-hand E-commerce platform. Based on the following case information, determine whether the buyer or the seller should be supported.
Case details: {`case_details`}
Return format:
```json
{"Reason": "Your reason", "Conclusion": "Buyer or Seller"}
```

---

### E.2. The prompt for juror persona awareness is as follows:

---

**Prompt E.2: Prompt for juror persona awareness**

You are an active member of an online community and are now serving as a juror for an E-commerce dispute. You must strictly adhere to the following persona while evaluating cases:

**[Personal Identity]**
Name: {name}
Demographics: {gender}, {age}, {status}

---

Professional Background: {role_description}

**[Psychological persona]**
Personality Traits: {traits}
Personal Interests: {interest}

**[Decision-making Philosophy]**
- **Juror Role:** You represent a specific segment of society. Your verdict should reflect your unique background, cognitive biases, and life experiences.
- **Consistency Rule:** When reviewing evidence, interpret the facts through the lens of your persona. For instance, a professional may focus on contractual details, while a regular consumer might prioritize fairness and emotional context.
- **Social Interaction:** In the upcoming collective discussion, maintain your persona's communication style (e.g., assertive, analytical, or empathetic) while interacting with the other 16 jurors.

## E.3. The prompts for the four stages of IV-CoT are provided below. The following prompt corresponds to Stage I:

**Prompt E.3: Prompt for IV-CoT in stage I**

Based on the case text, identify the buyer's and seller's demands and the reason for the dispute.
Case overview: {case_overview}
Product info: {product_info}
Buyer details: {buyer_details}
Seller details: {seller_details}
Please analyze:
1. What are the buyer's core demands?
2. What are the seller's core demands?
3. What is the focus of the dispute between the disputants?
Return format:
```json
{
    "buyer_core_claim": "...",
    "seller_core_claim": "...",
    "dispute_focus": "...",
}
```

## E.4. The prompt for selecting evidence in stage II is as follows:

**Prompt E.4: Prompt for IV-CoT in stage II to select evidence**

You are analyzing the case from the perspective of [{perspective}]. Select the piece of evidence whose visual content is most worth examining based on its accompanying text description.
Understood case: {stage1_output}
Analyzed evidence: {analyzed_text}
Description attached to evidence: {evidence_texts}
Based on the text description and the type of visual content, select the piece of evidence that is most likely to contain key visual information for in-depth analysis.
Return format:
```json
{
    "selected_evidence_id": "{perspective} evidence X",
```

```
    "reason": "reason for selecting",
}
```

## E.5. The prompt for perceiving evidence in stage II is as follows:

---

**Prompt E.5: Prompt for IV-CoT in stage II to perceive evidence**

You are analyzing the visual content of evidence from the perspective of [{perspective}].
Understood case: {stage1_output}
Product info: {product_info}
Evidence to analyze: {evidence_info}
Description attached to evidence: {evidence_texts}
Now analyze these visuals to find evidence in favor of [{perspective}]:
1. Carefully observe the key details in the picture/video
2. Return the path to the picture or video frame that contains the key information
3. Describe why the media is beneficial to [{perspective}]
4. Assess the strength of the evidence
Return format:
```json
{
  "visual_findings":[
    {
      "media_type": "image/video",
      "media_index": 0,
      "timestamp": "00:05",
      "description": "...",
      "benefit_analysis": "why it is beneficial to [{perspective}]",
      "importance": "...",
    }
  ],
  "evidence_summary": "...",
  "support_strength": "...",
  "is_sufficient": "true/false",
  "sufficiency_reason": "...",
}
```

---

## E.6. The prompt for analyzing evidence in stage III is as follows:

---

**Prompt E.6: Prompt for IV-CoT in stage III**

Conduct in-depth logical analysis based on key visual evidence provided by both parties.
Understood case: {stage1_output}
Buyer media paths: {buyer_media_paths}
Seller media paths: {seller_media_paths}
[Description]
- I have provided {len(buyer_images)} media items for the buyer and {len(seller_images)} media
items for the seller.
- Treat all media as a single list, with buyer media first and seller media last.
[Analysis Task] Please analyze the following content in depth:
1. **Root cause analysis of disputes**:

---

- What is the root cause of this dispute?
- Is it a product quality issue, a description mismatch, a service issue, or something else?
2. **Buyer's Dispute Position**:
- Why is the buyer dissatisfied? What are the specific demands?
- Does the buyer provide evidence (text + visual) to support their claims?
- What are the strengths and weaknesses of the buyer's claim?
3. **Seller's Dispute Position**:
- What is the seller's justification?
- Does the seller provide evidence (text + visual) to support its defense?
- What are the strengths and weaknesses of the seller's claim?
4. **Conflict Focus**:
- Where is the core point of disagreement between the two sides?
- Are there any contradictions in the evidence from both sides?
- What key information can be seen from visual evidence?
Return format:
```json
{
    "dispute_root_cause": "root cause for dispute",
    "buyer_position":{
        "main_complaint": "...",
        "demands": ...,
        "key_evidence": ["key_evidence1", "key_evidence2"],
        "visual_support": "whether the visual evidence supports the buyer",
        "strengths": ["strength1", "strength2"],
        "weaknesses": ["weakness1", "weakness2"],
    },
    "seller_position":{
        "main_defense": "...",
        "dcounter_arguments": "...",
        "key_evidence": ["key_evidence1", "key_evidence2"],
        "visual_support": "whether the visual evidence supports the seller",
        "strengths": ["strength1", "strength2"],
        "weaknesses": ["weakness1", "weakness2"],
    },
    "conflict_focus":{
        "core_disagreement": "...",
        "dcounter_arguments": "...",
        "evidence_contradictions": ["evidence_contradiction1", "evidence_contradiction2"],
        "visual_evidence_insights": "key information found from visual evidence",
        "critical_facts": ["critical_fact1", "critical_fact2"],
    },
}
```

**E.7. The prompt for making a final verdict in Stage IV is as follows:**

**Prompt E.7: Prompt for IV-CoT in stage IV**

You are juror #{agent_id}. It is now time to make a final ruling on this second-hand E-commerce transaction dispute.
Social consensus: {social_consensus}
Atmosphere of discussion: {mean_field}

Verdicts from the jurors you followed in the previous round: {agents_and_verdicts}
Buyer's claim: {buyer_claim}
Seller's claim: {seller_claim}
Analysis of buyer's evidence: {buyer_evidence_analysis}
Analysis of seller's evidence: {seller_evidence_analysis}
In-depth analysis: {stage3_output}
[Judgment Rules]
- Only after the buyer receives the goods, the goods seriously do not meet the seller's description, or there are serious quality problems, or the seller deliberately conceals the truth, etc., the buyer can return the goods for a refund with conclusive evidence.
- If the product matches the seller's description and the seller has fulfilled its obligation to inform the buyer and has not deliberately concealed defects and quality problems, the buyer cannot return the product for a refund unless there are special circumstances.
- Compared with product display images, the product description or product quality inspection report in the chat history is a more important basis for judging the status of the product.
- It is normal for second-hand goods to have certain defects or traces of use, unless there are major quality problems in the product, or the defects and traces of use are obvious but the seller has not informed them, otherwise the description of the product such as almost brand new does not constitute a problem of inconsistency in the description
- If the seller has displayed the product information before the transaction and provided the visual information of the product in the chat history, it proves that the buyer has recognized and understood the product and its description when placing the order, and then applies for a return for relevant reasons after receiving the goods.
- If the customer does not carefully read the product display and chat history and the return request (regardless of whether the product description is detailed or not), it means that the product received by the customer is seriously different from the expectation, and the buyer's request should be rejected unless necessary.
- Carefully check the reasonableness of the evidence and arguments of the buyer and seller to determine whether the buyer and seller have fabricated the evidence from the video pictures. Comprehensive consideration of evidence, taking into account possible evidence and negligence of both the buyer and seller.
- The dispute discussion between the buyer and the seller should be based on the status of the seller shipping and the buyer receiving the goods. The buyer should keep the condition of the goods before unpacking them, and after unpacking the express delivery, there are serious traces or visual evidence after using the purchased goods to examine the reasonableness.
[Judgment requirements]
- Consider all available information comprehensively, including the mean field, social consensus, and neighboring jurors' perspectives.
- Give independent judgment (don't blindly follow your neighbors)
Return format:
```json
{
    "verdict":"buyer/seller",
    "reasoning": ["reason1", "reason2"],
}
```

### E.8. The prompt for generating collective verdict summary is as follows:

**Prompt E.8: Prompt for generating collective verdict summary**

You are an administrative secretary working for a second-hand E-commerce platform. Your task is to synthesize the ongoing jury discussion regarding a transaction dispute into a structured analytical report. You may already have summarized an earlier report containing previous comments, but new arguments have since emerged. You must generate an updated analytical report by integrating the previous report with the latest comments.
The report must include the following sections:

1. Key arguments and perspectives mentioned by the commenters.
2. An assessment of whether there is a heated or intense debate.
3. The current prevailing orientation of public opinion (who is being supported).
Case Details: {content}
New Juror Arguments: {comment}
Previous Report : {mf_text}

## E.9. The prompt for generating verdict guidelines in the Precedent Base Construction is as follows:

**Prompt E.9: Prompt for generating verdict guidelines**

You are a senior E-commerce dispute analyst. Your task is to distill 2 to 4 universal, reusable verdict rules or key points from the collective opinions of the general public jury.
Case Details: {core_dispute}, {claims}, {evidence_description}
Collective Opinions of the General Public Jury: {buyer_views}, {seller_views}
Final Verdict: {winner}
When analyzing the current case, please refer to the following rules summarized from similar historical cases, which will help you form a more consistent conclusion: {rules_text}
Based on all the "General Public Jury Opinions" provided above, and incorporating the "Reference Rules from Similar Historical Cases" (if available), please summarize 2 to 4 core, cross-case reusable verdict rules.
Output Requirements:
1. Rules should be universal judgment standards for a category of issues, rather than case-specific details, and must be concise and refined.
2. Rules should be stated from a third-party perspective. For example: "When a product has significant undisclosed defects, 'non-returnable' clauses are generally invalid."
3. The output must be in JSON format and contain only one reflection_result field, with its value being an array of 2 to 4 strings.

## E.10. The prompt for assigning metadata tags is as follows:

**Prompt E.10: Prompt for Metadata-Tag**

Based on the product information below, classify it into the most suitable subcategory.
Product Information: ${product_text}
Optional categories are as follows:
[Digital & Appliances]
- Mobile Phones: Includes mobile communication devices
- Computers: Computing devices
- Cameras: Imaging devices
- Major Appliances: Large household appliances
- Small Appliances: Small household appliances
- Smart Devices: Smart home devices
- Digital Accessories: Accessories for digital products
[Fashion & Bags]
- Clothing: Various types of clothing
- Footwear: Various types of footwear
- Bags & Luggage: Various bags and cases
- Hats: Various headwear
- Accessories: Clothing accessories
[Home & Lifestyle]
- Furniture & Home Décor: Large home furnishings

- Kitchen & Daily Essentials: Cooking and daily items
- Maternity & Baby Products: Infant and toddler products
- Pet Supplies: Pet-related goods
- Sports & Outdoors: Sports equipment
- Automotive Supplies: Automotive-related goods
- Collectibles & Entertainment: Collectible and entertainment items
 [Virtual & Services]
- Digital Resources: Downloadable digital resources
- Gaming Accounts & Services: Gaming-related services
- Agency Services: Various agency services
- Local Services: Offline services
- Tutorials & Courses: Educational resources
 [Other]
- Uncategorized Secondhand Items: Other items
 Requirements:
 1. Return only the name of the most suitable subcategory (e.g., Mobile Phones, Computers, etc.).
 2. You must select from the subcategories listed above; do not return a main category name.
 3. Do not include any explanations or additional text.
 4. If it cannot be classified into a specific subcategory, return 'Uncategorized Secondhand Items'.

