# OpenReview forum: "CyberJurors: A Multi-Agent Simulation Task for E-Commerce Disputes Verdict"
_ICML.cc/2026/Conference — ICML 2026 regular_

### Official Review · Reviewer_N1aJ · 2026-03-12

**Soundness:** 3
**Presentation:** 2
**Significance:** 3
**Originality:** 3
**Overall Recommendation:** 4
**Confidence:** 4

**Summary:**

This paper introduces a new task, E-commerce Dispute Verdict (EDV), aimed at automating jury-style decisions for second-hand e-commerce disputes with multi-round, multimodal evidence. To support this task, the paper presents VerdictBench, a benchmark of 6,000 cases with transaction metadata, chat histories, multimodal evidence, and jury-derived verdicts, with stratified splits by category and difficulty. On top of that benchmark, the authors propose CyberJurors, a multi-agent framework that combines an Individual Verdict Chain-of-Thought (IV-CoT) for structured evidence-focused reasoning with a Jury Consensus Verdict (JCV) module that simulates multi-round deliberation guided by precedent-derived verdict rules. The experiments report substantial gains over single LLM/MLLM baselines and prior court-simulation approaches, including 72.92% accuracy versus 66.73% for AgentCourt, along with improved alignment to human jury vote proportions.

**Compliance With Llm Reviewing Policy:**

Affirmed.

**Final Justification:**

Adjusted overall recommendation and soundness score after rebuttal mostly addressed the issues.

**Key Questions For Authors:**

-

**Limitations:**

yes

**Strengths And Weaknesses:**

The paper identifies a setting that is materially different from legal judgment benchmarks: evidence is multi-round, multimodal, and governed by informal platform norms rather than rigid statutes. That distinction is well motivated and, in my view, meaningful. The benchmark statistics also support the claim that this is a nontrivial setting: the dataset spans five top-level categories, has heavy multimodal content (14.1 images and 0.9 videos per case on average), and includes a difficulty notion based on 17-juror vote margins. A major concern is that the strongest baseline gap is partly confounded by modality and system design. The appendix explicitly states that text-only LLM baselines omit visual evidence entirely, and that court simulators are kept in their original text-only form without multimodal adaptation. That makes CyberJurors stronger in more than one dimension at once: it has multimodal access, structured prompting, multiple agents, consensus rounds, and precedent guidance. As a result, the reported gains do not isolate whether the benefit comes from the multi-agent jury simulation, the structured IV-CoT, the handcrafted verdict rules, or simply better access to the evidence.
The paper presents CyberJurors as a multi-agent reasoning framework, but a substantial part of the behavior appears to come from manually specified rules and precedent-derived norms. In particular, the final judgment prompt contains explicit decision rules about when returns should be granted, what evidence is prioritized, and how second-hand defects should be interpreted. That is not necessarily a flaw, but it weakens the claim that the gains primarily demonstrate emergent jury-style reasoning rather than a well-engineered domain-specific decision template. This point should be made much more explicitly.

---

> ### Author Rebuttal · Authors · 2026-03-28
>
> **【Q1】 The strongest baseline gap is partly confounded by modality and system design.**
>
> **【A1】** Thanks for speaking highly of the EDV scenario and VerdictBench. We also deeply appreciate your feedback on the ablation studies. To explicitly clarify whether the performance gains stem from modality access, handcrafted verdict rules, structured reasoning, multi-agent jury simulation, or precedents, we provide a more comprehensive ablation study (original Table 3, Page 7) on the validation set in Table 1. This allows for a more fine-grained understanding of each component.
>
> **It demonstrates that in E-commerce dispute scenarios, CyberJurors effectively supports the complex EDV task through the structured reasoning of IV-CoT and the multi-agent discussion of JCV, rather than relying solely on modality access or a handcrafted prompt.**
>
> **Table 1. Ablation analysis of each component**
> |Method|Accuracy|F1|
> |:-|:-|:-|
> |Baseline(T)|0.6176|0.6134|
> |Baseline(T,I,V)|0.5416|0.5433|
> |+Rule Prompt|0.5876 (+0.0460)|0.5887|
> |+SR-CoT|0.6406 (+0.0530)|0.6416|
> |+IV-CoT|0.6734 (+0.0328)|0.6788|
> |+Jury|0.7018 (+0.0284)|0.7043|
> |+Precedent|0.7252 (+0.0234)|0.7196|
>
> ---
> > **【Q1.1】 Modality Confound.**
>
> - In our manuscript (Table 2, Page 6), we note a phenomenon: *"current MLLMs yield an average accuracy of 52.98%, which is notably inferior to the 55.78% achieved by text-only LLMs."*
> - The results of modality ablation in Table 1 align with the above phenomenon: in the multi-round, multimodal, and long-evidence-chain scenario of VerdictBench, *simply adding multimodal evidence does not naturally improve performance.*
> - To further verify whether this phenomenon persists across other models, we compare the performance of different  MLLMs under both text-only and multimodal conditions on the test set, as shown in Table 2. We also provide a multimodal adaptation for the text-based court simulator. The results reveal a consistent trend across different models: *access to visual evidence does not directly yield a performance advantage.* Conversely, when the visual evidence is removed, the accuracy and F1 metrics of these models actually improve.
>
> **Table 2. Performance of different baselines across different modalities**
> ||Modality(T)|Modality(T, I, V)|Difference|
> |:-|:-:|:-:|:-:|
> |Gemini-2.5-Flash-Lite|0.5925|0.5292|**-0.0633**|
> |Gemini-3.1-Flash-Lite|0.6968|0.6354|**-0.0614**|
> |Doubao-Seed-1.6|0.6369|0.5626|**-0.0743**|
> |AgentCourt| 0.6254 |0.6075|**-0.0179**|
>
> **The above phenomenon is because current MLLMs cannot perform fine-grained perception of key visual clues within ultra-long contexts and struggle to extract critical information from redundant multimodal evidence, leading to degraded accuracy. Therefore,  CyberJurors' significant performance is definitely not *simply better access to the evidence*.** Rather, through the structured task decomposition of IV-CoT, it effectively extracts key clues from redundant evidence, overcoming the interference caused by visual noise. Furthermore, by mitigating individual bias through JCV, CyberJurors further pushes accuracy and successfully replicates the voting distribution of the 17 real human jurors.
>
> ---
> > **【Q1.2】Demonstrate that gains come from the reasoning framework rather than the manually formulated rule prompt.**
>
> **【A1.2】** To more clearly distinguish the accuracy gains brought by the rule prompt, the structured reasoning, and multi-agent collaboration, we add an ablation study on the Rule Prompt in Table 1.
>
> - Building upon the baseline (Prompt in Appendix D.1), adding the verdict rule prompt (Appendix D.7) yields **a 4.6% accuracy improvement, demonstrating the limited contribution of the rule prompt.**
> - IV-CoT further improves accuracy by 8.58% (SR-CoT contributes 5.3%, and the "Select-Perceive" iterative clues grounding strategy contributes 3.28%). As shown in Appendix D.3-D.6, IV-CoT is designed purely to maintain the stage-wise reasoning pipeline for task decomposition, and its prompts are functional rather than heavily handcrafted. **This decisively proves that gains of IV-CoT stem not from a well-engineered decision template, but from the structured decomposition of the EDV task and the fine-grained perception of key clues.**
> - JCV provides an additional 5.18% accuracy improvement. **This gain is completely independent of the prompt design, illustrating the advantages of jury discussion (2.84%) and precedents (2.34%) in mitigating individual bias.** Additionally, we would like to clarify the positioning of CyberJurors' multi-agent framework. The design of JCV is not merely to push accuracy further; more importantly, it aims to replicate the voting distribution of the 17 actual human jurors in  E-commerce platforms (Page 1).
>
> Thanks for the effort in improving the quality of our work and for affirming the application of EDV. In the revised manuscript, we would explicitly describe the relative contributions of modality access, verdict rules, IV-CoT, and JCV.

---

> > ### Author Rebuttal · Reviewer_N1aJ · 2026-04-03
> >
> > As the 4.6% accuracy improvement is the second largest contributor, I would not characterize it as a “limited contribution.” Rather, it indicates that the rule prompt constitutes a meaningful part of the overall performance gains and should be described accordingly.
> > The ablation study is sequential rather than factorial, but it nevertheless provides useful evidence that both structured reasoning and the multi-agent component contribute non-trivially to performance. This clarification improves the transparency of the system and partially addresses my earlier concern regarding the attribution of gains across components.
> > I also appreciate the additional modality analysis. The observation that multimodal inputs do not automatically improve performance in this setting helps clarify that the improvements are not merely due to expanded evidence access, but are linked to how the evidence is processed. This strengthens the argument that structured reasoning plays an important role.
> > Overall, the rebuttal improves the paper by better disentangling the contributions of rules, reasoning, and multi-agent deliberation, and by making the system design more explicit.
> > I will adjust my score.

---

> > > ### Author Response · Authors · 2026-04-05
> > >
> > > We sincerely appreciate your recognition that both IV-CoT and JCV contribute non-trivially to the overall performance gains. We are also grateful for your positive assessment of the additional modality analysis. Your valuable comments have greatly improved the transparency and clarity of CyberJurors. We will carefully incorporate them to further strengthen the revised version, and we will revise our characterization of the rule prompt as making a "limited contribution." Thank you again for your careful consideration and rigorous views.

---

### Official Review · Reviewer_JJeg · 2026-03-13

**Soundness:** 3
**Presentation:** 3
**Significance:** 3
**Originality:** 3
**Overall Recommendation:** 4
**Confidence:** 3

**Summary:**

The paper introduces the E-commerce Dispute Verdict (EDV) task, which aims to automate the resolution of online transaction disputes—a process that currently relies on slow, crowdsourced human jury panels. Unlike legal dispute resolution, EDV requires reasoning over noisy, multimodal evidence chains (text, images, and videos) submitted across multiple rounds by buyers and sellers, guided by informal transaction conventions rather than formal legal codes. The paper provides VerdictBench, a benchmark of 6,000 real-world e-commerce dispute cases containing transaction logs, multi-round multimodal evidence, and ground-truth verdicts from 17-person juries. The paper also proposes CyberJurors, a multi-agent framework with two core components: Individual Verdict Chain-of-Thought (IV-CoT), which decomposes each juror’s reasoning into four stages (focus extraction, iterative clues grounding, adversarial analysis, and final verdict), and Jury Consensus Verdict (JCV), which simulates multi-round deliberation among heterogeneous jurors.

**Compliance With Llm Reviewing Policy:**

Affirmed.

**Final Justification:**

The rebuttal satisfactorily addressed my concerns, and I maintain my recommendation of weak accept.

**Key Questions For Authors:**

Please see Weaknesses

**Limitations:**

Yes

**Strengths And Weaknesses:**

**Strengths**

1. Identifying a genuine gap between existing legal dispute resolution research and the practical needs of e-commerce platforms. The EDV task is meaningfully distinct from legal verdict tasks along several dimensions. One of the main contributions, in my view, is VerdictBench, which represents a substantial contribution: a dataset of 6,000 real-world cases with authentic multi-round evidence chains, ground-truth verdicts from 17-person juries, and rich multimodal content that future researchers can build upon. The construction pipeline (including consistency checks, semantic alignment, privacy anonymization, and stratified partitioning) and the thorough statistical analysis of the dataset’s properties are also well done.
2. The IV-CoT's decomposition into focus extraction, iterative clues grounding, adversarial analysis, and verdict is not just a generic chain-of-thought , it directly targets the identified bottlenecks of existing approaches.

3. The experiments go well beyond reporting aggregate accuracy numbers. Evaluating across five distinct baseline categories (closed/open LLMs, closed/open MLLMs, and court simulators), providing a good picture of how different model families perform on EDV.

**Weaknesses**

1. Heavy reliance on a single (and relatively weak) backbone model limits the generalizability of conclusions. All CyberJurors experiments use Gemini-2.5-Flash-Lite-Nothinking as the core engine, which is notably one of the weaker MLLMs in the baseline comparison . While this choice is justified on efficiency grounds ,  simulating 17 jurors over multiple rounds is expensive — it raises questions about whether the framework's gains are partly compensating for the backbone's limitations rather than demonstrating broadly transferable design principles. Would IV-CoT and JCV still yield comparable improvements when built on a stronger backbone like Gemini-3-Pro or GPT-5.2?

2. I wonder how would this framework go against/integrated into Generative social choice framework [Fish et al.]?  I was wondering whether you can say improve CyberJurors by feeding into a Generative social choice setup?

 [Fish et al.] Generative Social Choice

---

> ### Author Rebuttal · Authors · 2026-03-27
>
> > **【Q1】 Backbone Generalizability: Does the framework generalize to stronger models? It is unclear if IV-CoT and JCV provide transferable improvements or merely compensate for the weak baseline.**
>
> 【A1】 We sincerely thank the reviewer for raising this critical question regarding the generalizability of IV-CoT and JCV across different backbones, which significantly helps strengthen the persuasiveness of CyberJurors. To demonstrate that we provide broadly transferable design principles rather than merely compensating for a weak baseline, we conduct new ablation studies using a stronger baseline: Gemini-3.1-Flash-Lite-Preview.
>
> **Table 1. Component Ablation on a Stronger Baseline**
> |Method|Accuracy|Weighted F1|Macro F1|
> |:-| :-| :-| :-|
> |Baseline|0.6886|0.6934|0.6858|
> |+IV-CoT|0.7394|0.7412|0.7253|
> |+JCV|0.7651|0.7663|0.7512|
>
> **The results in Table 1 demonstrate that IV-CoT and JCV consistently deliver robust performance gains when deployed on more powerful backbone models.**
>
> Compared to Gemini-3.1-Flash-Lite-Preview, incorporating the IV-CoT module yields a 5.08% improvement in accuracy, while the JCV module provides an additional 2.57% boost. **These results, which are consistent with our original ablation studies, strongly demonstrate that CyberJurors is not merely a simple compensation for the weak model's limitations. Instead, it serves as a transferable design principle for enhancing complex decision-making.** By equipping a single model with the structured reasoning of IV-CoT and the multi-agent consensus mechanism of JCV, CyberJurors paves the way for building a fair, robust, and reliable verdict system.
>
> ---
> > **【Q2】How would Cyberjurors go against / be integrated into Generative Social Choice?**
>
> 【A2】We sincerely appreciate the inspiring suggestion! CyberJurors shares a fundamental alignment with Generative Social Choice in addressing "how to forge collective decisions from diverse individual preferences." This perspective helps us significantly better position CyberJurors within broader collective decision-making paradigms.
>
> **We fully agree that Generative Social Choice offers a valuable theoretical perspective for open-ended collective decision-making.** It fuses the rigor of social choice theory with the flexibility and power of generative AI to reach collective answers in a scalable and principled way. It ensures that every diverse voice is not just heard, but proportionally and perfectly represented in the final slate.
>
> **Following your inspiration, we view CyberJurors and Generative Social Choice as a highly promising direction for future integration.** Specifically,
> - By aggregating the verdict results and reasons of numerous jurors across various cases, the framework can extract the jurors' verdict preferences;
> - IV-CoT-based generative queries produce verdict views that satisfy the preferences of jurors;
> - Discriminative queries then identify n/k  jurors satisfied with these views;
> - The above process iterates until a representative slate of k views satisfying the preferences of n jurors is generated.
>
> **Advantages**: This design not only 1) breaks through the limitations of black-and-white binary verdicts by generating compromise mediation plans (e.g., proportional refunds), but also 2) leverages the sampling mechanism to reduce the computational cost of the jury discussion. This facilitates scaling to larger jury sizes and ensures that diverse, reasonable views are fairly represented in the final slate. These align with the research vision for the  EDV and VerdictBench.
>
> **However, applying this integrated framework in practice faces two major challenges**: 1) Generative Social Choice assumes the LLM is an oracle that can precisely generate new alternatives and predict agents’ preferences. However, the data scarcity of E-commerce disputes in pre-training corpora leaves current models with an inherent bias (see Fig. 8(a), Page 18), challenging the assumption of 'perfect oracle'. Although IV-CoT has made notable progress, it necessitates the JCV to mitigate individual biases. 2) Generative queries generate views that maximize the utility of a subset of participants based on their preferences. Consequently, extracting their preferences requires gathering comprehensive verdict histories from a massive pool of jurors, which demands significantly more time and data collection efforts to extend VerdictBench.
>
> **As research on the EDV task deepens and VerdictBench expands, we believe CyberJurors will serve as a pioneering application of Generative Social Choice in the EDV domain.** We will incorporate the aforementioned discussion into the revised manuscript. Thank you again for sharing the visionary theoretical paradigm. We look forward to further discussions with you.

---

> > ### Author Rebuttal · Reviewer_JJeg · 2026-04-03
> >
> > Thank you for the new experiments and I appreciate the discussion related to generative social choice. I would like to maintain my judgement and my current score of weak accept for the paper.

---

> > > ### Author Response · Authors · 2026-04-05
> > >
> > > Thank you very much for your thoughtful suggestion and for carefully reading our rebuttal. We sincerely appreciate your positive comments and the discussion on generative social choice.

---

### Official Review · Reviewer_BbRC · 2026-03-13

**Soundness:** 3
**Presentation:** 3
**Significance:** 3
**Originality:** 3
**Overall Recommendation:** 5
**Confidence:** 3

**Summary:**

The paper introduces CyberJurors, a multi-agent framework designed to automate e-commerce dispute resolution. The system leverages LLMs acting as jurors who analyze multimodal evidence and engage in multi-round deliberations to reach a consensus. A primary contribution is the VerdictBench dataset, a large-scale collection of real-world disputes. The authors use mechanisms such as the Individual Verdict Chain-of-Thought and deliberation protocols inspired by social choice and opinion dynamics to mitigate individual LLM biases and improve overall verdict accuracy.

**Compliance With Llm Reviewing Policy:**

Affirmed.

**Final Justification:**

The author has addressed the concerned during rebuttal phase, I would like to raise the score to accept.

**Key Questions For Authors:**

- What is the computational cost (e.g., API calls, tokens, wall-clock time) of running the full K=17 juror, multi-round deliberation pipeline per case? How does this compare to the marginal accuracy gain over simpler baselines (e.g., single-agent or 3-agent)?
- The accuracy differences between configurations in your ablation studies (e.g., Table 2, Table 3) appear modest in some cases. What would be confidence intervals or significance tests to confirm that the observed improvements are statistically reliable and not artifacts of variance?
- How does the framework perform when applied to data from different jurisdictions, languages, or dispute types not represented in the current dataset?

**Limitations:**

Yes

**Strengths And Weaknesses:**

Strengths:
- The paper addresses automated dispute resolution, which has significant utility for e-commerce platforms.
- The VerdictBench dataset is a valuable addition to the field, providing a necessary benchmark for multimodal reasoning and dispute resolution.
- The use of structured deliberation drawing on social choice and opinion dynamics represents a more sophisticated approach than simple voting, helping to effectively surface and correct individual LLM biases.

Weaknesses:
- While the application is new, the underlying multi-agent mechanisms (debate, voting, consensus) are established techniques. The theoretical framing is present but could be more deeply analyzed. For example, there is no formal convergence guarantee or characterization specific to this setting.
- While some ablation studies are mentioned, a more granular analysis of the specific contribution of different components (e.g., the impact of the specific personas assigned to jurors, the necessity of the foreman role) would strengthen the paper.
- Given that the task involves subjective judgment (the paper itself notes 80.3% mean human accuracy), a human evaluation comparing system verdicts against expert judgments or a user study assessing the perceived fairness and quality of generated verdict explanations would significantly strengthen the paper.
- Running K=17 juror agents over T rounds implies substantial API costs and latency. The paper does not provide a cost or runtime analysis, which is important for practical deployment considerations.

---

> ### Author Rebuttal · Authors · 2026-03-28
>
> > **【Q1】Token Comparison & Jurors' Number.**
>
> 【A1】 Thanks for the insightful suggestions. Table 1 provides the token budget of CyberJurors evaluated on 100 randomly sampled cases. The results demonstrate that:
>
> - **IV-CoT achieves superior accuracy over strong baselines, Gemini-3-pro, with a significantly lower token budget.**
> - JCV is not designed solely to improve 5.18% accuracy. Rather, it aims to replicate the actual voting distribution of the 17 real human jurors in the VerdictBench (Page 1),  demonstrating the value of multi-agent research. **JCV should be viewed as a necessary choice for modeling real-world verdict dynamics, rather than a simple alternative to a single model.**
>
> **Table 1. Token Comparison on 100 cases**
> ||Accuracy|Token|
> |:-|:-|:-|
> |Gemini-3-Pro|0.6354|4,120,022|
> |1 juror with IV-CoT|0.6734|2,862,538|
> |17 jurors with JCV|0.7252|49,746,582|
>
> Furthermore, Table 2  provides an ablation study varying the number of agents. **As the number of jurors increases, the accuracy exhibits an upward trend, reaching a stable and saturating performance at the 17-juror scale** (N=17 aligns with the "crowdsourced jury" mechanism in real-world platforms).
>
> **Table 2. Ablation Study on the Jurors' Number**
> |Number|1|3|7|11|17|25|
> |:-|:-|:-|:-|:-|:-|:-|
> |Acc|0.6734|0.7079|0.7154|0.7201|0.7252|0.7319|
>
> For more details, please refer to our 【A1】 & 【A2】 to **Reviewer vAWV** and token analysis on Appendix C.4.
>
> ---
> > **【Q2】Significance tests on the observed improvements in Ablation Study (Page 7, Table 3).**
>
> 【A2】 Thanks for the valuable feedback regarding stability. Although each module consistently improves at least 2% accuracy, we agree that rigorous statistical testing is necessary. Thus, we repeat five independent runs per ablation setting and conduct statistical analyses from two perspectives.
>
> **Table 3. Stability Analysis & Paired-samples t-test for Module Ablations  (5 Runs)**
> |Method |P-value|t-statistic|
> |:-|:-|:-|
> |baseline|0.5705|-|
> |SR-CoT|0.7832|11.0297|
> |IV-CoT|0.7640|4.3637|
> |Jury|0.8392|6.3500|
> |Precedent|0.9098|3.6770|
>
> **Results prove that the incremental improvements of CyberJurors are statistically significant, highly stable, and not artifacts of variance.**
> - **Intra-Module Stability**: Consistent with the stability analysis (Appendix C.5, Page 19), Cochran‘s Q tests across the 5 runs yield p > 0.05 for all settings. **The results fail to reject the null hypothesis, meaning there is no statistically significant difference among the 5 independent runs.**
> - **Inter-Module Significance**: To verify the significance of the incremental gains, we conduct a Paired-samples t-test between adjacent modules. **Under a significance level of $\alpha=0.05$, all calculated t-statistics are larger than the critical value of 2.132, proving that the module's improvements are statistically significant.**
>
> ---
> > **【Q3】The performance applied to different scenarios.**
>
> 【A3】 Thanks for the valuable feedback regarding the cross-domain generalization performance of CyberJurors. To address this, we conduct experiments on the *Knowledge Question Answering* task from the *LawBench*.
>
> The task serves as an excellent testbed for evaluating a model's ability to apply legal knowledge and perform complex reasoning. It involves many complex domains, such as transnational trade and government subsidies. It requires the model to carefully distinguish subtle differences among options and make professional judgments based on legal knowledge.
>
> **Table 4. Cross-domain Generalization Performance**
> |Method|Accuracy|
> |:-|:-|
> |Baseline|0.6860|
> |CyberJurors |0.7220|
>
> CyberJurors achieves an accuracy of 72.2%, realizing a 3.6% improvement over the Gemini-2.5-Flash-Lite. **Although CyberJurors is specifically designed for EDV, the result demonstrates that its core capabilities—structured reasoning and multi-agent consensus verdict—are not limited to a single domain. They can effectively transfer to the broader legal knowledge question-answering domain.**
>
> ---
>
> > **【Q4】User study to assess the quality of generated verdict explanations.**
>
> 【A4】 To evaluate the persuasiveness of CyberJurors' explanations, we conduct a user study where 10 annotators assess 100 randomly sampled cases for "consistency" and "rationality" against the GT real verdict reasoning. The evaluation adopts a 10-point Likert scale (where 1 indicates completely inconsistent/irrational, and 10 indicates highly consistent and perfectly logical).
>
> **Table 5. User study on the Verdict Quality**
> |Annotator|**Average**|1|2|3|4|5|6|7|8|9|10|
> |:-:|:-:|:-:|:-:|:-:|:-:|:-:|:-:|:-:|:-:|:-:|:-:|
> |Score|**8.26**|8.70|7.96|8.18|7.88|8.40|8.64|7.95|8.91|7.75|8.23|
>
> The ten annotators give high scores for these generated explanations, reaching a strong average score of 8.26. **These results demonstrate that CyberJurors not only achieves an objectively high accuracy, but also the subjective verdict explanations it generates possess high quality and rationality.**

---

> > ### Author Rebuttal · Reviewer_BbRC · 2026-04-04
> >
> > Thanks for the response and explanation. I would like to raise the score to accept.

---

> > > ### Author Response · Authors · 2026-04-05
> > >
> > > Thank you very much for carefully reading our rebuttal. We are encouraged that our clarification has addressed your concerns and led to a more positive evaluation of CyberJurors. We will carefully incorporate your suggestions into the revised version. Thank you again for your time and support.

---

### Official Review · Reviewer_vAWV · 2026-03-13

**Soundness:** 3
**Presentation:** 3
**Significance:** 2
**Originality:** 3
**Overall Recommendation:** 3
**Confidence:** 4

**Summary:**

The paper introduces a novel task called E-commerce Dispute Verdict, which shifts the focus of LLM-based judicial simulation from formal legal proceedings to the informal, multimodal, and highly dynamic realm of online shopping disputes. The authors aim to examine a notable context where dispute resolution relies heavily on fragmented multimodal evidence and flexible transaction rules rather than rigid legal statutes. To support this task, the authors introduce VerdictBench, a large-scale dataset comprising 6,000 real-world e-commerce dispute cases. Furthermore, they propose CyberJurors, a multi-agent framework that combines an Individual Verdict Chain-of-Thought for fine-grained multimodal evidence grounding with a Jury Consensus Verdict mechanism to simulate multi-round jury discussions guided by historical precedents. Extensive experiments demonstrate that CyberJurors significantly outperforms existing LLMs, MLLMs, and prior court simulators.

**Compliance With Llm Reviewing Policy:**

Affirmed.

**Key Questions For Authors:**

1. While the application scenario is interesting and practically valuable, the paper aligns better with an application conference, such as ACL or EMNLP.

2. The proposed CyberJurors framework utilizes 17 agents engaging in multi-round discussions, which consumes significantly more inference-time compute than a standard zero-shot baseline. Could the authors provide a fair comparison where the single-model baseline is given an equivalent compute budget?

3. The system design, particularly the choice of exactly 17 agents and the specific social network topology, appears somewhat arbitrary. Could the authors provide detailed ablation studies on the number of agents?

**Limitations:**

yes

**Strengths And Weaknesses:**

Strengths:

1. This paper presents a novel and valuable task as well as dataset, bridging a gap where previous datasets were mostly text-only and focused on formal law.

2. The paper provides comprehensive experiments, including comparisons with a wide array of state-of-the-art LLMs and MLLMs, ablation studies, and token consumption analysis.

Weaknesses:

1. The computational cost is high. The multi-round, multimodal evidence across 17 agents still requires nearly 50M input tokens for just 100 cases. So it is not very surprising that the CyberJurors can outperform existing LLMs and MLLMs.

2. This paper focuses on a specific application without sufficient novelty from the machine learning algorithm perspective. So I think it is more suitable for ACL or CVPR.

---

> ### Author Rebuttal · Authors · 2026-03-28
>
> > **【Q1】 Provide a fair comparison of compute budget.**
>
> 【A1】 Thanks for the insightful suggestion. To ensure a fairer comparison, we compare the performance of the single-agent IV-CoT against strong baselines under equivalent compute budgets. Accordingly, we optimize the results presentation in Appendix C.4 (Page 18), and supplement it with a new lightweight ablation experiment.
>
> **Table 1. Performance and Token Comparison across Different Models**
> ||Accuracy|Token|
> |:-|:-|:-|
> |Gemini-2.5-Flash-Lite|0.5500|1,037,520|
> |Gemini-3-Pro|0.6354|4,120,022|
> |1 juror with IV-CoT|0.6734|2,862,538|
> |IV-CoT-Lite|0.6600|1,881,823|
> |17 jurors with JCV|0.7252|49,746,582|
>
> **Under a lower token budget than the strong baselines,  a single juror with IV-CoT achieves the optimal performance. Furthermore, the design of "17 jurors with JCV" not only improves accuracy but, more importantly, replicates the real-world voting distribution of the 17 human jurors (Page 1), demonstrating the value of multi-agent research.**
> - Compared to Gemini-3-Pro, IV-CoT achieves a +3.80% Acc. gain while reducing token consumption by 30.5%. This stems from IV-CoT's structured decomposition of the EDV task, which, as the Reviewer notes, "directly targets the identified bottlenecks of existing approaches" rather than merely relying on a larger token budget.
> - A lightweight version, IV-CoT-Lite (which retains our core innovations while streamlining secondary inputs), consumes only ~1.8x the tokens of Gemini-2.5-Flash-Lite but achieves a significant 11.0% improvement in accuracy.
> - **Clarification on JCV**:  JCV is not designed solely to mitigate individual bias or push accuracy. Importantly, it aims to simulate the "crowdsourced jury" mechanism in E-commerce platforms and to replicate the voting distribution of the 17 actual human jurors (Page 1).  JCV should be viewed as a necessary architectural choice for modeling real-world verdict dynamics rather than a simple alternative to a single-model setup.
>
> ---
> > **【Q2】 The choice of exactly 17 agents appears somewhat arbitrary. Provide detailed ablation studies on the number of agents?**
>
> 【A2】 Thanks for the attention to the scientific rigor behind our jury size setting (N=17).
>
> **Table 2. Ablation Study on the Number of Jurors**
> |Number of jurors| Accuracy|
> |:-:|:-|
> |1|0.6734|
> |3|0.7079|
> |7|0.7154|
> |11|0.7201|
> |17|0.7252|
> |25|0.7319|
> - Task Formulation: N=17 in JCV is not an arbitrary choice. **As briefly mentioned in 【A1】, JCV is designed to replicate the actual voting distribution of the 17 human jurors.** This provides a rigorous baseline and a clear methodological reference for future research.
> - Empirical Findings: To further substantiate this choice, we provide an ablation study varying the number of agents in Table 2. **It demonstrates a consistent upward trend: accuracy steadily increases as the number of agents scales up, reaching a stable and saturating performance at the 17-juror scale.**
>
> ---
> > **【Q3】While the application scenario is interesting and practically valuable,  the paper aligns better with an application conference.**
>
> 【A3】Thanks for acknowledging the practical value of our work. Regarding the conference fit, we respectfully argue that this paper perfectly aligns with the scope of ICML.
>
> - **The core contribution of CyberJurors goes far beyond a specific domain application. Rather, it introduces a novel task formulation, a rigorous benchmark, and a generalized framework for the ML community.** EDV and VerdictBench serve as realistic testbeds for exploring core ML challenges, including fine-grained multimodal perception, ultra-long context reasoning, and multi-agent dynamics, which have been highly praised by multiple reviewers. Besides, CyberJurors addresses two general ML bottlenecks: IV-CoT tackles content perception and causal reasoning within redundant, multi-round, and multimodal contexts; JCV models the multi-agent consensus-building process to mitigate individual bias.  CyberJurors also achieves generalizable performance gains in other scenarios (see Reviewer BbRC【A3】).
> - **CyberJurors deeply aligns with active research directions within the ICML community**,  including Multimodal CoT [Video-of-Thought, ICML 2024], Evidence-based Reasoning [DEFAME, ICML 2025], and Multi-agent System [AutoML-Agent, ICML 2025; Improving Factuality, ICML 2024].
> - **ICML states that *"it is globally renowned for publishing cutting-edge research used in ... important application areas,"* and it explicitly includes the *Applications* Area in its submission tracks.** As you kindly note, CyberJurors is *"interesting and practically valuable,"* providing a high-quality environment for evaluating advanced multi-agent systems, making it a perfect fit for this specific track at ICML.
>
> Given our strong contributions and deep alignment with ML community, we respectfully request the reviewer to reconsider the positioning of CyberJurors.

---

> > ### Author Rebuttal · Reviewer_vAWV · 2026-04-05
> >
> > Thanks for the response.
> >
> > After reading the response, although I do not have any follow-up questions, the two weaknesses still hold from my perspective. So I decided to keep my original weak reject.

---

> > > ### Author Response · Authors · 2026-04-05
> > >
> > > We thank you for acknowledging our rebuttal. However, given that we have provided extensive empirical results and detailed clarifications addressing your two original weaknesses, **we respectfully request that you elaborate on which specific aspects of these weaknesses still hold from your perspective.** Pointing out the specific aspects in our rebuttal would greatly facilitate a constructive scientific discussion.
> > >
> > > To reiterate our rebuttal regarding your two primary concerns:
> > >
> > > > **Token Consumption**
> > >
> > > **CyberJurors does not simply rely on scaling up token consumption. In fact, our single-agent IV-CoT is significantly more efficient than strong baselines.**
> > > - Under a lower token budget, a single juror with IV-CoT achieves optimal performance compared to Gemini-3-Pro.
> > > ||Accuracy|Token|
> > > |:-:|:-|:-|
> > > |Gemini-3-Pro|0.6354|4,120,022|
> > > |1 juror with IV-CoT|0.6734|2,862,538|
> > > - To provide a strictly fair comparison, our IV-CoT-Lite consumes only ~1.8x the tokens of Gemini-2.5-Flash-Lite, but achieves a massive 11.0% absolute improvement in accuracy.
> > > ||Accuracy|Token|
> > > |:-:|:-|:-|
> > > |Gemini-2.5-Flash-Lite|0.5500|1,037,520|
> > > |1 juror with IV-CoT-Lite|0.6600|1,881,823|
> > > - JCV further yields a 5.18% improvement in accuracy, highlighting the value of multi-agent system research. More importantly, **JCV should be viewed as a necessary architectural choice for modeling real-world verdict dynamics rather than a simple alternative to a single-model setup.**
> > >
> > > **Just as the computation demands of LLMs/MLLMs are exponentially higher than traditional models—yet their development is an unstoppable and essential trend—token consumption should not be a definitive barrier to advancing multi-agent systems, which govern the performance upper bound of complex reasoning tasks.**
> > >
> > > > **The positioning of CyberJurors**
> > >
> > > We respectfully disagree with the assessment regarding the paper's positioning.
> > >
> > >
> > > CyberJurors goes far beyond a specific domain application, strictly aligning with the *Applications -> Social Sciences* track. By introducing a novel task (EDV), a rigorous benchmark (VerdictBench), and a generalized multi-agent framework, **it provides the broader ML community with a highly challenging, real-world testbed for core challenges: fine-grained multimodal perception, ultra-long context reasoning, and complex social decision-making.**
> > >
> > > As explicitly stated on the ICML Website, it encompasses "*cutting-edge research on all aspects of machine learning used in closely related areas... as well as important application areas*." **As ICML represents a diverse and inclusive community, we believe providing realistic testbeds like VerdictBench is crucial. It bridges theoretical ML with real-world scenarios, directly supporting the ongoing efforts of researchers, entrepreneurs, engineers, and students in this rapidly growing field.**
> > >
> > > ```
> > >                                                About the Conference
> > > ICML is globally renowned for presenting and publishing cutting-edge research on all aspects of machine learning used in closely related areas like artificial intelligence, statistics and data science, as well as important application areas such as machine vision, computational biology, speech recognition, and robotics.
> > >
> > > ICML is one of the fastest growing artificial intelligence conferences in the world. Participants at ICML span a wide range of backgrounds, from academic and industrial researchers, to entrepreneurs and engineers, to graduate students and postdocs.
> > > ```
> > >
> > > We kindly urge you to point out any specific shortcomings in the rebuttal provided above. We remain fully open to discussion and hope to address your remaining concerns directly.

---

### Decision · Program_Chairs · 2026-04-30

**Decision:**

Accept (regular)

**Comment:**

The paper introduces CyberJurors, a multi-agent framework designed to automate e-commerce dispute resolution. The studied problem is very important and relevant, and has significant utility for e-commerce platforms. Most of the concerns are well addressed during rebuttals, so I would like to recommend accept on this paper.